# A rapid aureochrome opto-switch enables diatom acclimation to dynamic light

Huan Zhang [1,2], Xiaofeng Xiong[1], Kangning Guo [1], Mengyuan Zheng [3,4,5], Tianjun Cao[1,2], Yuqing Yang[1], Jiaojiao Song[1], Jie Cen[1], Jiahuan Zhang[1], Yanyou Jiang [1], Shan Feng[6,7], Lijin Tian [3,4,5] & Xiaobo Li [1,2] ✉

Diatoms often outnumber other eukaryotic algae in the oceans, especially in coastal environments characterized by frequent fluctuations in light intensity. The identities and operational mechanisms of regulatory factors governing diatom acclimation to high light stress remain largely elusive. Here, we identified the AUREO1c protein from the coastal diatom *Phaeodactylum tricornutum* as a crucial regulator of non-photochemical quenching (NPQ), a photoprotective mechanism that dissipates excess energy as heat. AUREO1c detects light stress using a light-oxygen-voltage (LOV) domain and directly activates the expression of target genes, including *LI818* genes that encode NPQ effector proteins, via its bZIP DNA-binding domain. In comparison to a kinase-mediated pathway reported in the freshwater green alga *Chlamydomonas reinhardtii*, the AUREO1c pathway exhibits a faster response and enables accumulation of LI818 transcript and protein levels to comparable degrees between continuous high-light and fluctuating-light treatments. We propose that the AUREO1c-LI818 pathway contributes to the resilience of diatoms under dynamic light conditions.

Photosynthetic organisms provide oxygen, food, and other resources that sustain the human society and the entire biosphere. A striking pattern exists in the distribution of eukaryotic photosynthetic groups. While terrestrial areas are predominantly covered by green plants (Viridiplantae) that harbor a plastid that likely originated from an engulfed cyanobacterium, marine environments are dominated by algae with complex plastids that may have evolved from eukaryotic endosymbionts[1]. Despite the presence of many aquatic species within Viridiplantae (the green algae), photosynthetic eukaryotes in oceans are mainly represented by diatoms, accounting for 20% of global primary productivity[2,3]. In addition to serving as a major carbon sink, diatoms represent a sustainable source of many nutraceuticals, such as the highly valued carotenoid fucoxanthin[4]. Therefore, comprehending the molecular mechanisms underlying the ecological success of diatoms bears significant biogeochemical and economic implications. Moreover, diatoms have unique photosynthetic properties, like their superior ability to harness blue-green light common in underwater settings[5–8] (Supplementary Fig. 1). If integrated into food crops, this could help fill the green gap in their light utilization spectrum.

In their natural environment, diatoms are exposed not only to a different light spectrum compared to green plants, but also to different dynamics of light input. They therefore require distinct strategies for light acclimation. Specifically, although light provides energy for photosynthetic organisms, high levels of light can cause damage[9]. Acclimation to high light thus entails the activation of various mechanisms to prevent such damage.

[1]Research Center for Industries of the Future, Key Laboratory of Growth Regulation and Translational Research of Zhejiang Province, School of Life Sciences, Westlake University, Hangzhou, China. [2]Institute of Biology, Westlake Institute for Advanced Study, Hangzhou, China. [3]Key Laboratory of Photobiology, Institute of Botany, Chinese Academy of Sciences, Beijing, China. [4]University of Chinese Academy of Sciences, Beijing, China. [5]China National Botanical Garden, Beijing, China. [6]Mass Spectrometry & Metabolomics Core Facility, The Biomedical Research Core Facility, Center for Research Equipment and Facilities, Westlake University, Hangzhou, China. [7]Key Laboratory of Structural Biology of Zhejiang Province, School of Life Sciences, Westlake University, Hangzhou, China. ✉e-mail: lixiaobo@westlake.edu.cn

A broadly-conserved strategy for preventing light damage involves dissipation of excess light energy as heat, a process known as the non-photochemical quenching (NPQ) of chlorophyll fluorescence[9–11]. The short-term component of NPQ in photosynthetic eukaryotes, energy-dependent quenching (qE)[12], involves the concerted actions of de-epoxidized xanthophyll carotenoids and effector proteins[11,13]. The effector proteins of qE are proteins in the PSBS or LI818 families[14,15]. Notably, LI818 proteins, initially discovered in green algae, are not found in the closely-related vascular plants, but are present in the more distantly-related diatoms[14,16–23]. They are generally referred to as LHCSR proteins in green algae and LHCX proteins in diatoms.

Estuarine and coastal diatoms continually experience minute-scale light fluctuations due to water motion, with light intensities reaching 2000 μmol photons m$^{-2}$ s$^{-1}$ at mid-day[24,25]. They show stronger induction of NPQ relative to algae living in other habitats; furthermore, in laboratory settings mimicking fluctuating light, estuarine

and coastal diatoms show greater resistance to high light than the other algae[26–30]. Previous work identified the possession of dual pathways of xanthophyll de-epoxidation in diatoms (in contrast to the single pathway in green plants) to be a cause of the strong induction of NPQ in diatoms. It was also shown that the additional pathway in diatoms acts with fast kinetics[31–34]. Additionally, several studies have demonstrated that certain members in the *LHCX* family are rapidly transcriptionally activated by the onset of high light, in some cases within 5 min[22,35–38]; and the importance of LHCX members in diatom NPQ has been demonstrated[17,20,21,23]. However, the mechanisms by which high light regulates the expression of specific LHCX isoforms remain unknown.

Recent studies in land plants, green algae, and diatoms all suggest that both chloroplast signals and photoreceptor-mediated light signaling contribute to high light acclimation[11,39–47]. For example, in the land plant *Arabidopsis thaliana*, the two phototropins, PHOT1

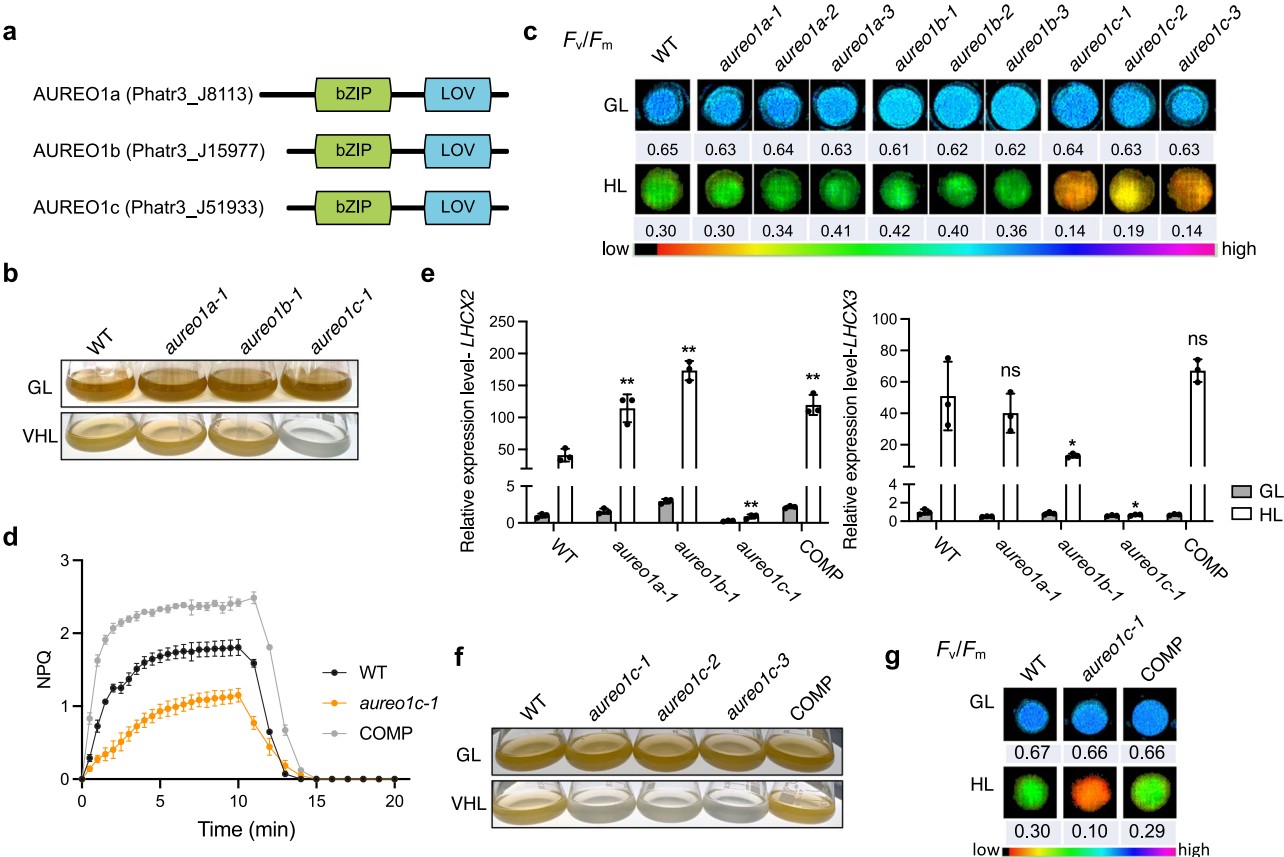

**Fig. 1 | AUREO1c is required for *LHCX2/3* gene expression, NPQ development, and maintenance of photosynthetic performance under high light. a** Diagrams of the three aureochromes in *P. tricornutum*. **b** Photographs of cultures of wild-type (WT) and aureochrome mutants grown under our standard light conditions (GL, 40 μmol photons m$^{-2}$ s$^{-1}$ of white light) and treated for 2 days under very high light (VHL; 1200 μmol photons m$^{-2}$ s$^{-1}$ of white light). **c** Maximal quantum yields of photosystem II (PSII) for WT cells and aureochrome mutants, after growth under GL and 3 days of treatment under high light (HL; 550 μmol photons m$^{-2}$ s$^{-1}$ of white light), measured as the ratio of variable chlorophyll fluorescence ($F_v$) to the maximum fluorescence ($F_m$) after dark acclimation (see "Methods"). The $F_v/F_m$ ratios are shown as false-color images, with values indicated below each image. **d** NPQ phenotypes of WT, the *aureo1c-1* mutant, and the COMP strain treated for 6 h under 900 μmol photons m$^{-2}$ s$^{-1}$ of white light. Data are presented as mean values ± standard deviations (SD). **e** Relative transcript abundance of *LHCX2* and *LHCX3* in WT, aureochrome mutants, and the *aureo1c-1* complemented line (COMP; expressing the wild-type *AUREO1c* gene in *aureo1c-1* mutant cells) normalized to the average of the WT-GL samples. The measurements were performed by RT-qPCR. Data are presented as mean values ± SD. The GL and HL growth conditions were the

same as those in (**c**). HL treatment continued for 1 h before cells were collected for analysis. The expression level of *LHCX2/3* in aureochrome mutants and COMP line under high light conditions was compared with that of WT cells using a two-sided *t*-test. \**p* < 0.05; \*\**p* < 0.01; ns, not significant. For *LHCX2*, the *p* values for the comparisons were 0.0061 for *aureo1a-1* vs. WT, 0.0002 for *aureo1b-1* vs. WT, 0.002 for *aureo1c-1* vs. WT, and 0.002 for COMP vs. WT, respectively; for *LHCX3*, the *p* values were 0.493 for *aureo1a-1* vs. WT, 0.040 for *aureo1b-1* vs. WT, 0.016 for *aureo1c-1* vs. WT, and 0.291 for COMP vs. WT, respectively. **f** Phenotypic rescue of *aureo1c-1* by complementation with the wild-type *AUREO1c* gene (the COMP strain), observed as appearance under VHL (same condition with VHL in (**b**)). In addition, photographs of the cultures of the two other mutant lines of *AUREO1c* are shown. **g** Phenotypic rescue of *aureo1c-1* by genetic complementation based on the $F_v/F_m$ ratio. The GL and HL conditions were the same as those in (**c**). Uncropped images with additional replicates (independent cultures or independent mutant lines) for (**c**, **g**) are provided in Supplementary Fig. 6a, b. For (**b**–**g**), three independent cultures were used for the measurements. The experiment was repeated three times independently with consistent results and a representative result is shown. Source data are provided as a Source Data file.

(aka. NPH1) and PHOT2 (aka. NPL1), each comprising two blue light-sensing LOV domains and a serine/threonine kinase domain, have overlapping functions in modulating blue light-mediated phototropism and chloroplast accumulation under low light. However, PHOT2 exhibits lower light sensitivity, i.e., it requires a higher light intensity to reach its maximum function[44]. Pertinent to this, PHOT2 possesses a specialized role in mediating the photoprotective chloroplast avoidance response under light stress, which is not shared by PHOT1[43,44]. In the green alga *Chlamydomonas reinhardtii*, the singular phototropin (PHOT1) has been recently identified as a key player in activating *LHCSR3* expression under high light conditions[45]. For diatoms, while tremendous progress has been made in the understanding of the chloroplast signals[48–50], a photoreceptor that is essential for acclimation and upregulation of *LHCX* genes under high light has not been reported. Previous photoreceptor studies in diatoms mostly addressed gene expression or other responses to blue or red light with an intensity below 150 μmol photons m$^{-2}$ s$^{-1}$ [51–57]. However, exposure to higher light intensities could potentially manifest a pronounced phenotype, such as a visible bleaching phenotype[45], in mutants of photoreceptors that play a key role in the regulation of photoprotection.

In this study, we took advantage of the recently developed CRISPR/Cas9 technologies in the model coastal diatom *Phaeodactylum tricornutum*[58] to disrupt its candidate regulatory genes and investigate their roles in *LHCX* gene expression and acclimation to light stress. Our efforts uncovered the blue light receptor PtAUREO1c to be an important and specific regulator of genes involved in photoprotection and photosynthesis under high light. By combining biochemical, cell biological, and multi-omic approaches, we found that AUREO1c perceives blue light and instantaneously activates the transcription of two *LHCX* genes, in a manner quantitatively dependent on the light intensity. Finally, we demonstrated the importance of this pathway in acclimation to prolonged high light and rapidly fluctuating light.

## Results

### The photoreceptor AUREO1c is required for *LHCX2/3* gene expression, NPQ development, and maintenance of photosynthetic performance under high light

To understand how diatoms respond to high-light stress, we initially aimed to identify the relevant LHCX family members involved in acclimation to high light in *P. tricornutum*. Among its four *LHCX* isoforms, although *LHCX2* and *LHCX3* are upregulated under high light[39,59], *P. tricornutum* cells harboring a mutation in *LHCX2* or silenced *LHCX3* expression exhibit only modestly lower NPQ than wild-type cells[20,60]. To investigate whether LHCX2 and LHCX3 can functionally compensate each other, we generated and phenotyped their single and double mutants (see Supplementary Data 1 for all primers) under light stress. We found that double, but not single, mutants deficient in *LHCX2* and *LHCX3*, were bleached under very high light (1200 μmol photons m$^{-2}$ s$^{-1}$ of white light; referred to as "VHL"; see Supplementary Fig. 2 and Supplementary Data 2 for culture conditions throughout the paper), suggesting that LHCX2 and LHCX3 act redundantly in maintaining cell fitness under light stress (Supplementary Figs. 3a and 4a–c). Using lower light intensities that did not cause bleaching, we observed that the double mutants show decreased NPQ and a lower fraction of variable chlorophyll fluorescence ($F_v/F_m$) ratio compared to wild-type cells (Supplementary Fig. 3b, c); lower $F_v/F_m$ could arise from oxidative damage of photosystem II (PSII)[61,62]. These findings indicate that LHCX2 and LHCX3 play a critical function in photoprotection.

We then sought for *P. tricornutum* photoreceptors important for its high light acclimation. Among the multiple photoreceptors encoded by the *P. tricornutum* genome, three proteins (AUREO1a, AUREO1b, and AUREO1c) belong to the stramenopile-specific, blue light-sensing aureochrome family[63,64]. Aureochromes contain a single C-terminal LOV domain and an N-terminal bZIP transcriptional factor domain, and can mediate light-induced transcriptional regulation[65–69]. We prioritized this

family because of three reasons. First, in a large-scale transcriptomics study, AUREO1b and AUREO1c appeared in the same co-expression modules with genes implicated in photosynthesis (chlorophyll biosynthesis genes, "lightsteelblue1" module) and photoprotection (*LHCX3*, "blue" module) respectively[70]. Secondly, previous studies demonstrated the importance of blue, but not red, light signals in the induction of *LHCX2/3* expression[39,59,71]. Thirdly, photoreceptors exhibit conformational changes upon exposure to light and revert to their original conformation upon exposure to darkness and the reversion kinetics are negatively correlated with their light sensitivity[72]. In an in vitro study, recombinant AUREO1c proteins showed faster recovery than AUREO1a proteins after blue light excitation, suggesting that AUREO1c has lower light sensitivity and may play a physiological role in high light sensing[73].

We therefore mutagenized AUREO1a, AUREO1b, and AUREO1c and selected three mutants for each gene displaying unambiguous sequencing results near the guide region for further characterizations (Fig. 1a and Supplementary Figs. 4d–f and 5; "Methods"). In these selected mutants, we observed a lack of mosaicism (evident homozygosity) near the guide region. This could be the result of identical repair outcomes following biallelic cutting or gene conversion, a process regularly observed in *P. tricornutum*[58,74]. Unique insertions or deletions (InDels) were found in each of the three mutant lines, according to our sequencing results. These InDels are anticipated to induce a frameshift during translation, leading to functional inactivation of the protein (Supplementary Fig. 4). The single-nucleotide polymorphisms (SNPs) observed in wild type could be found in some of the mutants, and their disappearance in other mutants may have resulted from mismatch repair associated with gene conversion[75] (Supplementary Fig. 5). All the mutants resembled wild type in appearance and physiological parameters when grown under our standard light conditions (referred to as "growth light" or "GL"; Fig. 1b, c).

Under VHL, *aureo1c-1* cells were bleached, while *aureo1a-1* and *aureo1b-1* mutants were not (Fig. 1b). Under lower light intensities, *aureo1c* mutants exhibited a lower $F_v/F_m$ ratio and NPQ (Fig. 1c, d). We conclude that *aureo1c-1*, but not *aureo1a-1* or *aureo1b-1*, resembled *lhcx2 lhcx3* double mutants (Supplementary Fig. 3) in high light sensitivity and AUREO1c is important for high light acclimation.

To delve deeper into the function of AUREO1c, we examined the performance of the *aureo1c-1* mutant under VHL using quantitative methods and over a time scale of hours to better simulate natural habitats. Wild type and the *aureo1c-1* mutant did not bear a major difference in pigment composition in GL (Supplementary Table 1). During the first day of VHL exposure, the cell concentration of the wild type doubled, whereas the *aureo1c-1* mutant exhibited no growth (Supplementary Fig. 7a). After a full day of exposure to VHL, a substantial loss in chlorophyll fluorescence was observed in most *aureo1c-1* cells, as evidenced by flow cytometry (Supplementary Fig. 7b–h). Concurrent HPLC measurements indicated an 8-fold decrease in chlorophyll *a* content in wild type but a 60-fold decrease in the *aureo1c-1* mutant, compared to the GL control cells (Supplementary Table 1). Following 2 days of VHL treatment, the pigment levels in the *aureo1c-1* cells approached our detection limit, rendering accurate quantification unfeasible. We decided to assess photosynthetic parameters within the initial 6 h of VHL treatment, during which the *aureo1c-1* mutant still retained chlorophyll levels comparable to the wild type (Supplementary Table 1). At this stage or earlier timepoints, we observed the mutant exhibiting a lower $F_v/F_m$ (noticeable as early as 2 h into the treatment), and NPQ (measured at 4 h) already (Supplementary Figs. 8 and 9), demonstrating that AUREO1c's role is likely important in the natural habitats, where VHL would not last longer than several hours.

We then investigated the mechanism how AUREO1c might impact NPQ. Our analysis of the expression of the four *LHCX* genes revealed that *aureo1c-1* deviated markedly from the wild type, showing a lower

accumulation of *LHCX2* and *LHCX3* transcripts under high light conditions (Fig. 1e and Supplementary Fig. 10). Because NPQ also depends on the activity of de-epoxidized xanthophyll carotenoids, we measured the xanthophyll de-epoxidation state (DES). Intriguingly, the *aureo1c-1* mutant exhibited a DES higher than wild type under two different light intensities tested (6 h in 1200 µmol photons m$^{-2}$ s$^{-1}$, Supplementary Table 1; 2 days in 900 µmol photons m$^{-2}$ s$^{-1}$, Supplementary Table 2). These results suggest that AUREO1c is specifically required for induction of *LHCX2* and *LHCX3* under high light. Intriguingly, the DES in the *aureo1c-1* mutant is elevated compared to that in wild type. A higher DES was also found in the *C. reinhardtii npq4* mutant[16] and this may be a common compensatory response to impaired NPQ.

To rule out potential off-target effects associated with the single guide RNA chosen for mutagenesis, we performed genetic complementation on one of the mutants disrupted in *AUREO1c* ("Methods"). The complemented line (COMP), expressing an AUREO1c-GFP fusion protein in the background of the *aureo1c-1* mutant, exhibited rescued *LHCX2/3* expression, NPQ, and $F_v/F_m$, and showed tolerance to very high light (Fig. 1d–g and Supplementary Figs. 6b, 7–9 and 11), confirming that the phenotypes of the *aureo1c-1* mutant were caused by the disruption of *AUREO1c*. We observe that the COMP line was even higher than wild type in NPQ (Fig. 1d and Supplementary Fig. 9), higher in *LHCX2/3* expression in certain conditions (Fig. 1e; "Dark to GL 10 min" samples in Supplementary Fig. 13), and lower in DES (Supplementary Tables 1 and 2). This apparent "over-complementation" could potentially be attributed to the use of the potent EF1α promoter, which drives the expression of AUREO1c-GFP in this line ("Methods").

Based on the above results, we conclude that AUREO1c is the key regulator required for the NPQ response and for survival of *P. tricornutum* after exposure to high light. Furthermore, it exerts its function (partially) through regulating the expression of *LHCX2/3* genes but not xanthophyll de-epoxidation.

## The contribution of AUREO1c to the accumulation of *LHCX2/3* transcripts increases with light intensity

In *A. thaliana*, the function of PHOT1 in modulating phototropism reaches saturation at light intensities above 10 µmol photons m$^{-2}$ s$^{-1}$. Conversely, the function of PHOT2 remains unsaturated at this intensity, aligning with its physiological role in regulating photoprotective responses under high light conditions[42,44]. To explore the effective light intensity range for AUREO1c to regulate *LHCX2/3* expression, we measured the expression of these two genes in wild type and the *aureo1c-1* mutant samples exposed to varying light intensities. In wild type, we observed a continuous increase in *LHCX2/3* expression with escalating light intensity up to the highest intensity investigated (640 µmol photons m$^{-2}$ s$^{-1}$ of white light). Furthermore, the ratio of *LHCX2/3* expression in wild type to that in the *aureo1c-1* mutant also showed a continuous increase within this range (Supplementary Fig. 12), instead of saturating at a low light intensity. This indicates that AUREO1c function indeed has a potential to serve as a quantitative indicator for the onset of high light stress.

Light intensity fluctuates during the day, and also changes during day-night transitions. In the case of *C. reinhardtii*, it has been observed that during dawn, despite the low light intensity, the transition from darkness to light also activates *CrLHCSR3* expression[76]; and this is partially dependent on CrPHOT1. Analogously, we found a similar response in *PtLHCX2* and *PtLHCX3*, which were expressed at a higher level when dark-acclimated cells were shifted to growth light, akin to *CrLHCSR3*; and the *aureo1c-1* mutant was partially blocked in this process (Supplementary Fig. 13). Moreover, the *aureo1c-1* mutant exhibits a deficiency in the upregulation of *PtLHCX2/3* during the transition of cells from darkness to high light, in addition to the previously observed deficiency during the shift from growth light to high

light (Fig. 1e and Supplementary Fig. 12). These findings suggest that AUREO1c functions in modulating *LHCX2/3* across various degrees of light intensity augmentation.

Subsequently, we evaluated *LHCX2/3* expression under a spectrum of blue light intensities after a period of incubation in darkness (Supplementary Fig. 14). For a comprehensive understanding, we deliberately incorporated light intensities as low as 5 µmol photons m$^{-2}$ s$^{-1}$, the minimal threshold our laboratory apparatus can provide, to investigate the "minimum light intensity" necessary to activate AUREO1c's role in modulating *LHCX2/3* expression. Our observations were multifold. Initially, we noted that *LHCX2/3* expression in the wild type augmented in parallel with blue light intensity. Transitioning from darkness to 5 µmol photons m$^{-2}$ s$^{-1}$ of blue light, *LHCX2* and *LHCX3* respectively exhibited approximately 7-fold and 4-fold increases in transcript abundance. This finding aligns with a prior study[54], although their documented upregulation was more pronounced (25-fold at 3.3 µmol photons m$^{-2}$ s$^{-1}$ of blue light) and this may be related to the different lengths of acclimation in dark (60 h in their study versus 12 h in this study). Secondly, we observed a progressive increase in the ratio of *LHCX2/3* expression in the wild type compared to the *aureo1c-1* mutant. Thirdly, even under a light intensity of 5 µmol photons m$^{-2}$ s$^{-1}$, the wild-type expression surpassed that of the *aureo1c-1* mutant by a factor of 2–3. However, the minimum intensity of blue light required to elicit an observable activation of AUREO1c function−specifically, as evidenced by a discernible difference in *LHCX2/3* expression from the wild type−remains undetermined.

We observed a few complications in the above experiments. First, it's important to note that regulators of *LHCX2/3* expression extend beyond just AUREO1c. These additional factors become particularly noticeable during transitions from darkness to growth light or high light ("Dark to GL 10 min" and "Dark to HL 10 min" samples in Supplementary Fig. 13), or to blue light (Supplementary Fig. 14). During these transitions, the *aureo1c-1* mutant also demonstrated a prominent induction of *LHCX2/3* relative to its own control in darkness. The AUREO1c-independent factors, which could possibly encompass the electron transport chain or other photoreceptors[39], remain to be conclusively identified and warrant further investigation. Secondly, for wild type, the *aureo1c-1* mutant, and the COMP line, cells switched from darkness to growth light 10 min prior to the quantification ("Dark to GL 10 min" samples in Supplementary Fig. 13) had a higher level of *LHCX2/3* transcript abundance than cells cultured in GL ("GL" samples). This outcome is not surprising because gene induction can be transient and relaxed after a period of time[39,40].

The aforementioned results, coupled with the presence of the bZIP domain, support the notion of AUREO1c functioning as both a blue light sensor and a transcription factor. Nevertheless, certain caveats exist: (1) The observed upregulation of *LHCX2/3* expression in conjunction with increasing light intensity via AUREO1c could potentially be attributed to the effect of amplified light energy, rather than light signaling; (2) The localization and DNA-binding capabilities of AUREO1c remain unexplored through experimental investigation. In the following sections, we address these considerations in greater detail.

## AUREO1c is a nucleus-localized sensor for blue light

To gain insights into the mechanism by which AUREO1c may regulate NPQ and *LHCX* gene expression, we first examined its subcellular localization. To do so, we took advantage of the COMP line that expresses the AUREO1c-GFP fusion protein. We found that the GFP fluorescence signal co-localized with DNA staining by 4′,6-diamidino-2-henylindole (DAPI) in cells grown under both growth light and high light. Control wild-type cells did not exhibit a visible signal in the GFP channel (Fig. 2a). These results indicate a nuclear localization of AUREO1c, consistent with its proposed role as a transcription factor.

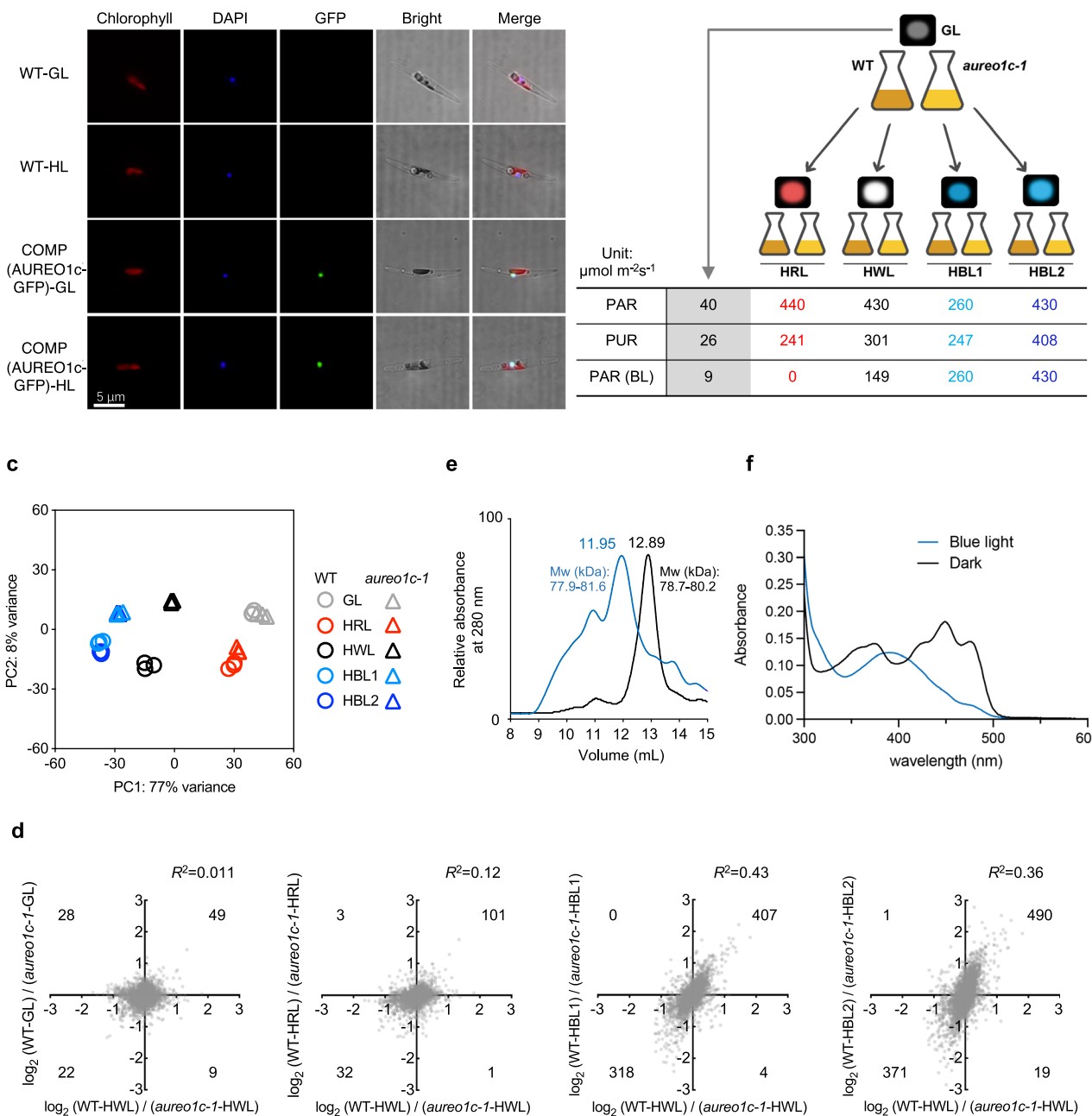

**Fig. 2 | AUREO1c is a nucleus-localized sensor for strong blue light.**
**a** Localization of AUREO1c-GFP protein under growth light (GL) and high light (HL). Wild-type (WT) cells lacking the transgene did not show detectable signal in the GFP channel, suggesting the signal for the COMP line is from the AUREO1c-GFP fusion protein. **b** Schematic showing conditions and treatments for transcriptome analysis. Three independent cultures were sampled each for WT and the *aureo1c-1* mutant. HWL, HRL, HBL stand for high white light, high red light, and high blue light respectively. The photosynthetic active radiation (PAR), photosynthetic usable radiation (PUR), and the blue light fraction of the PAR [PAR (BL)] for each condition were calculated and shown in the table. See "Methods", Supplementary Fig. 2 and Supplementary Data 2 for further details. **c** Principal component (PC) analysis of the transcriptomes of WT and the *aureo1c-1* mutant under various light conditions. Note that PC1 explains the bulk (77%) of the variation. **d** Correlation between the effects of the *aureo1c-1* mutation on transcriptomes under different conditions. The

*y*-axes show the log₂ values of the ratios between normalized expression levels in WT and mutant cells, under one of these four light regimes: GL, HRL, HBL1, and HBL2. The corresponding values under HWL condition are plotted along the *x*-axes. Each dot represents one gene. The squared Pearson's correlation coefficients ($R^2$) are indicated. All correlations had a two-sided *p* value < 0.0001. In each quadrant, the number of genes showing significant (FDR < 0.01; see "Methods") differences between WT and the mutant under both conditions is shown in the corner. **e** Size-exclusion chromatography of the recombinant proteins of AUREO1c, after dark incubation or blue light irradiation, followed by crosslinking. The molecular weight (Mw) labels are based on data in Supplementary Fig. 15. **f** Absorption spectra of recombinant AUREO1c proteins after dark incubation or blue light irradiation. For (**a**, **e**, **f**), the experiment was repeated three times independently, yielding similar results and a representative result is shown.

Moreover, our results suggest that light level does not affect its localization. Instead, it may alter its conformation, as reported previously for other aureochromes[77–79] and supported by the evidence presented below.

To understand whether the function of AUREO1c is blue light signal-associated, we compared the transcriptomes of the *aureo1c-1* mutant and wild-type cells under different light input conditions: growth light (GL), high white light (HWL), high red light (HRL), high

blue light 1 (HBL1) and high blue light 2 (HBL2) (Fig. 2b, Supplementary Fig. 2, and Supplementary Data 2). HBL2 was similar to HRL and HWL in photosynthetic active radiation (PAR) but higher in photosynthetically usable radiation (PUR) by 70% and 36%, respectively. HBL1 was additionally included to control for the actual photosynthetic energy intake; it was 18% lower than HWL and similar to HRL in PUR (Fig. 2b). Each of these conditions yields a distinct transcriptomic signature (Supplementary Data 3 and 4). However, principal component analysis (PCA) revealed major deviations of *aureo1c-1* mutant cells from wild-type cells under HWL, HBL1, and HBL2, but not under GL or HRL conditions (Fig. 2c). This analysis suggests that the deficiency in AUREO1c function mainly affects the cells' transcriptomic response to an increase in blue light, rather than total input light energy.

We further confirmed the blue-light specificity of AUREO1c by comparing the effect of *aureo1c-1* mutations on the transcriptome of cells grown under different light conditions (GL, HRL, HBL1, and HBL2) to the effect of *aureo1c-1* mutations on cells grown under HWL. We observed a positive correlation between transcriptomic changes under the two HBL conditions and HWL. In contrast, correlations between the effects on cells grown under GL or HRL and those grown under HWL were rather weak (Fig. 2d), despite that HRL and HBL1 were similar in PUR. Specifically, the *aureo1c-1* mutation caused consistent changes from wild type in the expression of more than 300 genes between HWL and HBL1/HBL2; "consistent" indicates a positive value in both the HWL and the HBL1/HBL2 axes or a negative value in both axes (Fig. 2d). By contrast, only around 100 or fewer genes showed a consistent change caused by the *aureo1c-1* mutation when HWL was compared to GL or HRL conditions (Fig. 2d and Supplementary Data 5). These results further support that AUREO1c orchestrates *P. tricornutum* gene expression responses to high light by sensing the strong blue light component, and validate that our datasets can be used to characterize the AUREO1c-dependent response to blue light.

Since AUREO1c mediates a response to blue light, we asked how blue light might impact the protein to allow regulation of its target genes. Previous work suggested distinct, protein-specific models of aureochrome activation. For instance, it was discovered that VfAUREO1 from the xanthophyte *Vaucheria frigida* transitions from a monomer to a dimer after blue light exposure, while PtAUREO1a remains predominantly dimeric in both dark and illuminated conditions[69,77,78,80,81]. We investigated the elution profiles of recombinant AUREO1c protein in size-exclusion chromatography (SEC) after a period of incubation in the dark or under blue light. The blue light-exposed sample eluted earlier than the dark-incubated sample (Fig. 2e). The absorption spectra of dark- or blue light-incubated AUREO1c proteins proved that the chromophore flavin mononucleotide (FMN) was embedded in the recombinant proteins that we obtained. Specifically, we detected a peak at 450 nm in the dark-incubated sample, attributable to non-covalently bound FMN, and a peak at 390 nm in the blue light-exposed sample, corresponding to the FMN-cysteine adduct[82] (Fig. 2f). Because earlier elution in SEC can be caused by either a larger size (associated with oligomerization) or a change in protein shape, we applied ultra high-performance liquid chromatography-native mass spectrometry (UHPLC−nMS) to measure the molecular weight of the protein samples. Both blue light-exposed and dark-incubated AUREO1c samples were determined to be predominantly dimerized (Supplementary Fig. 15). This indicates that AUREO1c can form a dimer under both conditions, a characteristic reminiscent of the behavior observed in AUREO1a[78]. The structural differences between dark-state and lit-state AUREO1c dimers await further investigations.

## AUREO1c regulates multiple processes related to photosynthesis and photoprotection, and can bind the promoters of *LHCX2* and *LHCX3*

Given its nuclear localization and the presence of a bZIP transcriptional regulatory domain, AUREO1c is likely to function as a transcriptional regulator. We therefore sought to identify its targets. The above transcriptomic analysis showed that HWL, HBL1 and HBL2 caused both AUREO1c-dependent and AUREO1c-independent changes from GL (Fig. 2c). To identify genes specifically regulated by AUREO1c, we selected genes that showed differential regulation under HWL relative to GL in wild-type cells and then asked whether those changes were disrupted in the *aureo1c-1* mutant ("Methods"). Gene ontology (GO) analysis showed that the AUREO1c-dependent set is enriched for genes involved in photosynthesis and chloroplast activities, whereas the AUREO1c-independent set is enriched for genes involved in other processes, such as nicotinamide adenine dinucleotide (NAD) binding (Fig. 3a and Supplementary Data 6).

As an alternative approach to characterizing AUREO1c-dependent genes, we specifically examined the expression of candidate genes that may play a role in sunlight energy harvesting and conversion under growth light or in NPQ under high light[6]. These include genes encoding components of the photosynthetic electron transport chain (ETC) and genes in the light-harvesting complex (LHC; including LHCX) family (Fig. 3b). Most of these genes were downregulated in wild-type cells upon exposure to HWL, but retained a higher expression level in the *aureo1c-1* mutant cells under the same conditions. A smaller number of genes were upregulated in wild-type cells in response to HWL compared to GL, but not in *aureo1c-1* mutant cells (Fig. 3b). This indicates that AUREO1c is important for both down- and up-regulation of high light-responsive genes.

We next investigated ETC and LHC genes in detail. Among ETC genes, we observed a specific effect on PSII. Under HWL, *aureo1c-1* mutant cells expressed higher levels of PSII reaction center subunits compared to wild-type cells (Fig. 3c), suggesting a role for AUREO1c in modulating the abundance of and potentially the ratios between different ETC complexes. Among LHC genes, we identified five different response groups, two of which depended on AUREO1c. Most of the *LHCF* genes (category I) were downregulated in wild-type cells under HWL, HBL1, and HBL2 compared to GL. *LHCX2*, *LHCX3*, *LHCR6*, *LHCR8*, and *LHCR10* (category II) were up-regulated under HWL, HBL1, and HBL2 relative to GL, consistent with previous reports[22,38,39,59] (Fig. 3d). Changes in the expression of genes in both of these categories were abolished or weakened in the *aureo1c-1* mutant cells, suggesting that their regulation by blue light depended on AUREO1c. Category III genes were induced by HBL1/HBL2 in wild-type cells, but in an AUREO1c-independent manner. Finally, *LHCF15* (category IV) and *LHCZ1* (category V) did not fall into these three clusters, consistent with previous expression clustering studies carried out in wild type[59,83].

The lower read counts for *LHCX2* and *LHCX3* in *aureo1c-1* mutant compared to wild-type cells is consistent with our RT-qPCR results presented above (Fig. 1e), providing a validation of the transcriptome datasets and confirming that *LHCX2* and *LHCX3* are regulated by AUREO1c at the transcriptional level.

*LHCX2* and *LHCX3* are among the most prominent downstream effectors of AUREO1c, based on the significant change in gene expression levels between wild-type and *aureo1c-1* mutant cells under HWL (Fig. 3b). They also show a significant difference in gene expression between HWL and GL in wild-type cells. These observations prompted us to ask whether *LHCX2* and *LHCX3* might be direct targets of AUREO1c. Because chromatin immunoprecipitation sequencing (ChIP-Seq) methods have not been established for transcription factors in *P. tricornutum*, we investigated whether AUREO1c could bind the promoter of *LHCX2* and *LHCX3* by performing DNA affinity purification sequencing (DAP-Seq)[84]. Enriched signals were observed at the promoters of these two genes, but not in the negative control sample performed using unfused GST proteins instead of GST-AUREO1c (Fig. 3e and Supplementary Fig. 16). We further confirmed the interactions between *LHCX2/3* promoters and AUREO1c by performing an electrophoretic mobility shift assay (EMSA). The probe containing the *LHCX2* or *LHCX3* promoter sequence exhibited

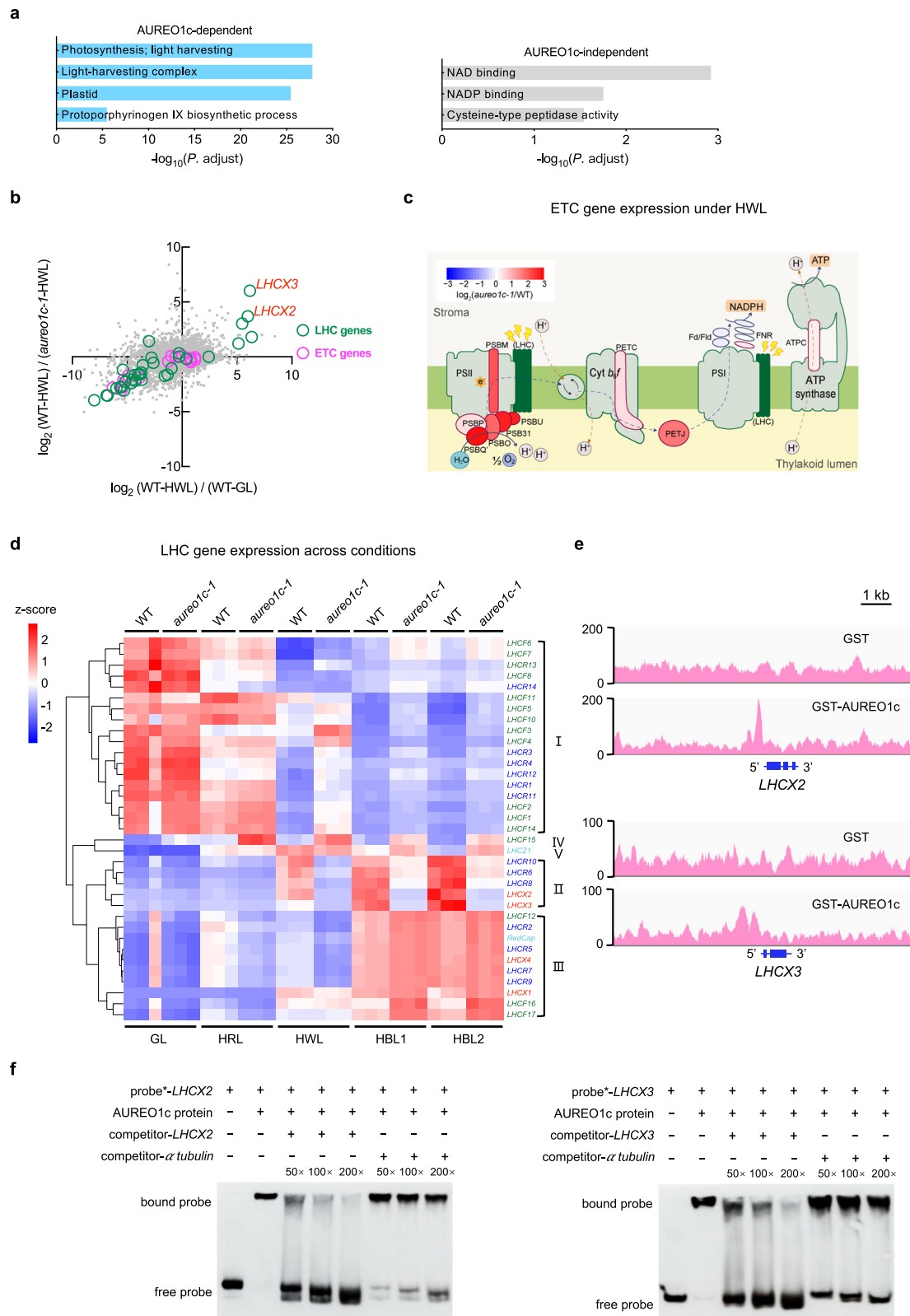

reduced mobility in the presence of AUREO1c (Fig. 3f). Adding a large excess of unlabeled probes reverted the effect on mobility of the labeled probes; in contrast, when the competition was performed using the promoter sequence of alpha tubulin, a gene not affected by the *aureo1c-1* mutation (Supplementary Data 4), most of the labeled probes were still in the upper band. These results demonstrate that AUREO1c can bind to the promoter of *LHCX2* and

*LHCX3* in a sequence-specific manner and may regulate their transcription directly.

## Rapid induction of *LHCX2/3* gene expression in *P. tricornutum* enables a strong response to fluctuating light

In the green alga *Chlamydomonas reinhardtii*, the high light-sensing phototropin protein (PHOT kinase) also contains a LOV domain and it

**Fig. 3 | AUREO1c regulates multiple processes related to photosynthesis and photoprotection, and can bind the promoters of *LHCX2* and *LHCX3*. a** Enriched gene ontology (GO) terms for genes regulated in HWL compared to GL in wild type (WT), separated into AUREO1c-dependent and AUREO1c-independent gene groups (see "Methods" for definitions). These GO terms had an adjusted *p* value (*p*. adjust) <0.05. The *p* values were obtained from the one-sided Fisher's exact test used in the clusterProfiler package, and Benjamini−Hochberg method was used to adjust for multiple comparisons (see "Methods"). The length of each bar is calculated based on its adjusted *p* value. **b** A scatterplot of all the genes in *P. tricornutum* genome showing their responses to HWL in WT cells and the effect of the *aureo1c-1* mutation on their expression. LHC family genes and nuclear genes encoding subunits of the photosynthetic electron transport chain (ETC) are highlighted. The data plotted were averages from three independent cultures. **c** Genes encoding ETC components color-coded to show relative expression levels. The ratio of the transcript abundance of ETC genes between *aureo1c-1* and WT under HWL is shown as a heatmap; red and blue indicate higher and lower expression in *aureo1c-1* compared to WT respectively. Many of the ETC subunits are encoded by the chloroplast genome and could not be detected in regular RNA-Seq experiments; these are shown as the gray background for each complex. Association between the light-harvesting complex (LHC) family members with photosystem II or photosystem I has not been entirely elucidated. They are shown as green rods here, with their

transcript abundance details in (**d**). Fld flavodoxin, FNR ferredoxin-NADP⁺ oxidoreductase, PS photosystem, Cyt *b₆f* cytochrome *b₆f* complex. Two *FLD* and four *FNR* gene isoforms were detected and plotted. The data plotted were averages from three independent cultures. **d** A heatmap showing the expression levels of the chlorophyll-binding protein-encoding genes in the LHC family. The genes were hierarchically clustered into five major categories (I−V). The expression level for each of the three independent cultures under each condition was plotted separately as a column. **e** DNA affinity purification sequencing (DAP-Seq) results showing enrichment of promoter regions of *LHCX2* and *LHCX3* in sequences bound to the AUREO1c protein. Note that different gene model annotations exist for LHCX2, and the Phatr2_54065 model shown in this panel is supported by our RNA-Seq and PCR validation results presented in Supplementary Fig. 16. This experiment was repeated twice independently with consistent results and a representative result is shown. **f** Electrophoretic mobility shift assays (EMSAs) showing the binding of recombinant AUREO1c to DNA probe harboring sequences from the promoters of *LHCX2* and *LHCX3* respectively. The competing unlabeled probes were applied in excess, as indicated (50−200x). An upper shift of the labeled probes suggests their binding to the AUREO1c protein. The experiment was repeated three times independently, yielding similar results and a representative result is shown. Source data are provided as a Source Data file.

transmits strong blue light signals to upregulation of *LHCSR* genes, by inhibiting a ubiquitin ligase complex that degrades the transcription factors required for *LHCSR* expression[85−87]. The direct regulation of *LHCX* expression by AUREO1c suggested that diatoms may be able to mount a faster response to high light. To test this idea, we compared the kinetics of *LI818* induction in diatoms and green algae using RT-qPCR over a time course during exposure to HBL. Interestingly, *PtLHCX2* and *PtLHCX3* transcript abundance was already statistically significantly elevated, relative to baseline, within 2 min of HBL exposure (Fig. 4a). By contrast, *CrLHCSR3.1* and *CrLHCSR3.2* required 6 min to reach significant induction; in this experiment, *C. reinhardtii* cells were grown photoautotrophically in the high-salt (HS) medium as *P. tricornutum* cells were (Fig. 4a). Besides the HS medium, *C. reinhardtii* can grow mixotrophically in the tris-acetate-phosphate (TAP) medium and acetate was found to repress *CrLHCSR3* expression[40]. However, the induction kinetics of *CrLHCSR3* transcript accumulation were consistent in TAP medium (Supplementary Fig. 17). The similar fold changes noted after 10 min of high light exposure between HS- and TAP-grown cells could be attributed to an elevated baseline expression in GL conditions in the HS medium (Supplementary Fig. 18), which is consistent with previous studies[40].

We hypothesized that a rapid induction of *LHCX2* and *LHCX3* expression in *P. tricornutum* may be critical in the context of the rapid, minute-scale, light fluctuations in estuarine or coastal habitats[24,25]. By contrast, the *C. reinhardtii* pathway may not enter a rapid accumulation phase for *LHCSR3* transcripts in each high light period. To test this hypothesis, we compared two light stress regimes: continuous high light treatment (CL) versus fluctuating light treatment (FL) with repeated cycles of 5-min high light followed by 5-min growth light. We asked whether the *P. tricornutum* kinetic pattern would allow significant accumulation of LHCX2/3 transcripts and proteins (and hence protection from light stress) under the FL regime. We performed RT-qPCR to compare *LI818* expression between comparable CL and FL treatments in both *C. reinhardtii* and *P. tricornutum*. Transcript abundances of *PtLHCX2* and *PtLHCX3* were similar after either FL or CL treatment, bearing a 50−75x upregulation from 0 to 30 min of treatment (Fig. 4b, c). The drop in *PtLHCX3* expression after 1 h of high light treatment is consistent with previous observations[39]. *CrLHCSR3.1* or *CrLHCSR3.2* was expressed to a much lower level in the FL regime compared to the CL regime in both the HS medium (Fig. 4c) and the TAP medium (Supplementary Fig. 19), at 30 and 60 min of treatment. It should be noted that in cells cultivated in the HS medium, the *LHCSR3* gene expression ceased to escalate after

60 min of CL treatment (Fig. 4c and Supplementary Fig. 20) whereas this cessation has not occurred at 60 min in the TAP medium (Supplementary Fig. 19). We further caution that the difference between FL and CL in *C. reinhardtii* is in part due to the large fold increase in the CL samples.

To confirm the functional relevance of the above results, we also examined LI818 protein abundance. We performed quantitative proteomics to measure the abundance of LI818 proteins in *C. reinhardtii* and *P. tricornutum* (Supplementary Data 7−9). We sampled wild-type cells grown under normal light conditions right before the transition to CL or FL (0 min), wild-type cells after 120 min of CL treatment, and FL cells that had undergone 120 min of high-light periods in total. LHCSR3 protein accumulation in *C. reinhardtii* was lower under FL compared to CL in HS medium (Fig. 4d), and the difference is more pronounced in TAP medium (Supplementary Fig. 21). In contrast, LHCX2 protein abundance increased to a similar degree under either CL or FL in *P. tricornutum* (Fig. 4d). Consistent with the above transcript abundance data, LHCX3 induction at 120 min of either CL or FL was limited. These results support the robustness of the *P. tricornutum* kinetic pattern when environmental light intensity fluctuates in minute-scale cycles. Conversely, the kinetic pattern in *C. reinhardtii* shows a less efficient induction of NPQ effectors under FL compared to CL. Corroborating these findings, we observed a lower NPQ in *C. reinhardtii* following FL treatment compared to CL treatment (Fig. 4e). Intriguingly, despite similar levels of LHCX2 and LHCX3 proteins, *P. tricornutum* displayed a higher efficiency in inducing NPQ under FL conditions compared to CL. This finding does not contradict our model, but does underscore the potential influence of various confounding factors that may affect NPQ. Further investigations are warranted to elucidate these effects.

Finally, for *P. tricornutum*, we additionally asked whether the ability to accumulate LHCX proteins under CL or FL depended on AUREO1c. To address this question, we compared protein accumulation between wild-type and *aureo1c-1* mutant cells at the same timepoint (120 min) under either CL or FL. The *aureo1c-1* mutant was unable to induce LHCX2 protein accumulation by either CL or FL treatment (Fig. 4d). For LHCX3, *aureo1c* mutations did not significantly impact protein abundance. However, the limited effect may be due to the fact that LHCX3 protein accumulation was already limited at 120 min even in wild-type cells. These results are consistent with the idea that AUREO1c is required for an optimal response to fluctuating light in *P. tricornutum*. Further supporting this idea, we found that the *aureo1c-1* mutant and *lhcx2 lhcx3* double

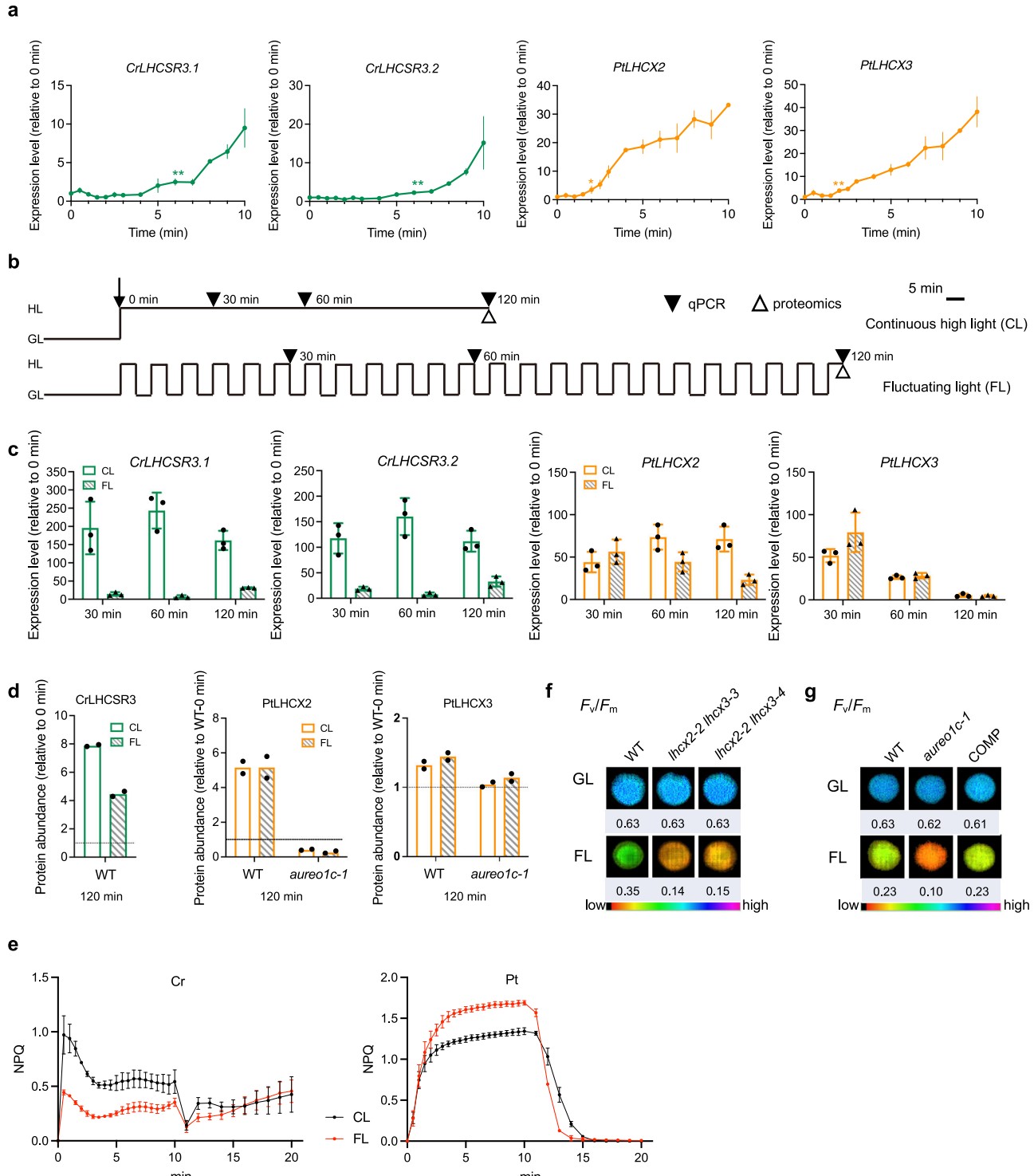

mutant in *P. tricornutum* showed sensitivity to prolonged growth in FL (Fig. 4f, g).

We conclude that the AUREO1c-dependent regulatory strategy used in *P. tricornutum* to respond to light stress increases the robustness of this organism under conditions of rapidly fluctuating light.

## Discussion

In our research, we initially discovered the critical role of AUREO1c in acclimating *P. tricornutum* cells to high light conditions and in initiating the expression of the NPQ effector genes *LHCX2* and *LHCX3*. We further analyzed the transcript abundance of *LHCX2* and *LHCX3* under various light treatments in both the wild type and an AUREO1c mutant (*aureo1c-1*), demonstrating that AUREO1c functions across a broad range of light intensities, notably exerting a stronger influence under more intense light conditions. We also conducted RNA-Seq under different colors and intensities of light and found that AUREO1c's function is contingent on blue light. Collectively, these data support a model where AUREO1c detects blue light signals and triggers high-level NPQ effector gene expression under high light conditions. Indeed, previous work, in both laboratory settings and environmental surveys, has shown that diatom AUREO1c proteins are highly expressed during the day[68,78], consistent with a role in photoprotection.

**Fig. 4 | AUREO1c rapidly induces *LHCX* gene expression upon high light and enables transcript and protein accumulation under rapidly fluctuating light.**
**a** Transcript abundance of *LI818* genes in wild-type *C. reinhardtii* (shown in green) and *P. tricornutum* (shown in orange brown) based on RT-qPCR at different time points after the switch from growth light (GL; 40 μmol photons m$^{-2}$ s$^{-1}$ of white light) to high blue light (260 μmol photons m$^{-2}$ s$^{-1}$). Three independent cultures were used for the quantification. Data are presented as mean values ± SD. The data were analyzed by using a one-sided unpaired *t*-test (*$p < 0.05$; **$p < 0.01$; and the *p* value corresponding to ** or * for *LHCSR3.1*, *LHCSR3.2*, *LHCX2* and *LHCX3* are 0.0024, 0.0048, 0.0183 and 0.00009 respectively). Correction for multiple comparisons was not performed. **b** Diagram of light regimes for *LI818* transcript abundance and LI818 protein abundance analysis. Cells were initially grown under GL and then switched to continuous high light (CL; 550 μmol photons m$^{-2}$ s$^{-1}$ of white light) or fluctuating light (FL; 550 μmol photons m$^{-2}$ s$^{-1}$ of white light alternating with 40 μmol photons m$^{-2}$ s$^{-1}$ of white light). **c** Transcript abundances of *LI818* genes in wild-type *C. reinhardtii* and *P. tricornutum*, after CL or FL treatments, relative to those under GL, measured by RT-qPCR. Samples from each condition receiving the same total length of high light treatment were compared. Three independent cultures were used for the quantification. Data are presented as mean values ± SD. **d** LI818 protein abundance in *C. reinhardtii* wild type (WT), *P.*

*tricornutum* WT and the *aureo1c-1* mutant, measured by proteomics. CrLHCSR3.1 and CrLHCSR3.2 are identical in amino acid sequence and are shown collectively as "CrLHCSR3" for *C. reinhardtii* WT, the data were normalized to the protein abundance of LHCSR3 under GL (shown as a dashed line). For *P. tricornutum* WT and the *aureo1c-1* mutant, the data were normalized to the protein abundance of LHCX2 or LHCX3 in WT under GL (shown as a dashed line). Two independent cultures were used for the quantification and the means are presented. This experiment was repeated twice independently with similar results and a representative result is shown. **e** The NPQ of *C. reinhardtii* and *P. tricornutum* after 2 h of CL and FL treatments. Three independent cultures were used for the quantification. Data are presented as mean values ± SD. **f** Maximal quantum yields of PSII ($F_v/F_m$) of wild type (WT), and two *lhcx2 lhcx3* mutants grown under fluctuating light (as the FL scheme in **b**) for 5 days. The $F_v/F_m$ ratios are shown as false-color images with values shown below. **g** Maximal quantum yields of PSII ($F_v/F_m$) of WT, the *aureo1c-1* mutant, and the COMP line, grown under fluctuating light (as FL in **b**) for 5 days. Uncropped images with additional replicates (independent cultures) for (**f**, **g**) are provided in Supplementary Fig. 6c, d. For (**a**, **c**, **e**–**g**), the experiment was repeated three times independently with similar results and a representative result is shown. Source data are provided as a Source Data file.

Our finding that AUREO1c acts by regulating the transcription of *LI818* genes (*LHCX2* and *LHCX3*) reveals unexpected convergence between diatoms and green algae, groups that are only distantly related to each other, having separated about two billion years ago[88]. Although both express blue-light sensors—PHOT kinases in green algae and aureochromes in diatoms—that contain LOV domains, the proteins differ in topology and in the way they activate downstream effects. PHOT proteins contain a kinase domain is C-terminal to the LOV domain, whereas aureochromes possess a bZIP domain is positioned N-terminal to the LOV domain. It is therefore remarkable that both species use of LOV family photoreceptors to control the expression of orthologous NPQ effector genes. The linking of the LOV domain with a bZIP domain in the diatom AUREO1c protein has streamlined the NPQ regulatory pathway, allowing AUREO1c to directly regulate *LHCX2/3* (Fig. 3e). This rewiring obviates the need for an additional regulatory step, and likely enables the rapid induction of LHCX2 and LHCX3 proteins in response to high light. This observation is consistent with a recent modeling study predicting that for the same input and output, molecular switch architecture based on direct activation (rather than derepression) allows faster induction of the output[89]. This dynamic regulation feature of diatoms likely contributes to their dominant occupancy of coastal habitats (Fig. 5). On the other hand, the green algal PHOT-LHCSR3 pathway may allow transcription factors to increase in abundance over time, leading to a faster-than-linear accumulation of *LHCSR3* transcripts; this rapid accumulation was observable within a span of 30 min for cells cultivated in the HS medium, and within 120 min for those grown in the TAP medium (Supplementary Figs. 19 and 20). This pattern is potentially advantageous when high light duration is longer than several minutes. Our study provides a model system for comparative characterizations of the different regulatory strategies.

We would like to underscore several precautionary points that could guide future research directions. Firstly, the breadth of light conditions we could test was confined by the limitations of our laboratory equipment. For a more comprehensive validation of our model, future research should consider investigating a wider array of light conditions. In particular, utilizing conditions like sinusoidal fluctuations may more closely mimic natural light environments[35]. Second, as revealed in this study (Fig. 4c and Supplementary Fig. 20) and previous studies on *CrLHCSR3.1*, *CrLHCSR3.2* (in the HS medium) and *PtLHCX3*, relaxation of gene induction occurs within 2 h[39,40,85]. To capture more detailed kinetics in future research, consideration could be given to incorporating more data collection points.

Another caveat in our study is that the AUREO1c-LHCX pathway operates in the context of pathways mediated by other photoreceptor

proteins, the chloroplast electron transport chain (ETC), reactive oxygen species, and/or other regulatory proteins[50,51,54,55], that together constitute the full light-stress response in *P. tricornutum*. For instance, the activation of *LHCX2/3* expression during dark-to-light transitions also occurs in the *aureo1c-1* mutant (Supplementary Figs. 13 and 14), indicating the presence of AUREO1c-independent factors to activate *LHCX2/3* expression. In addition, a previous study showed the presence of negative regulators of *LHCX2/3*, namely CPF1, a cryptochrome-type photoreceptor[54]. Moreover, a recent investigation in *C. reinhardtii* suggested that a low intracellular $CO_2$ level plays a crucial role in inducing *CrLHCSR3* expression[40]; although the conservation of this pathway in diatoms remains to be investigated. We acknowledge that some of these additional factors could also contribute to the kinetic features of diatom acclimation to a dynamic light climate or even interact with the AUREO1c pathway. To disentangle the regulatory and kinetic roles of each factor, future kinetic studies could be conducted in the context of genetic or pharmacological perturbations, or under varying light conditions, including transitioning from a dark environment.

From a broader perspective, the AUREO1c light switch has significant potential in synthetic circuit design. For land plants, accelerated induction and relaxation of photoprotection can confer enhanced protection and minimize energy losses[90,91]. The demand for rapid high light-based switches may also increase in the context of a changing climate[92]. In addition, photoreceptors have been extensively employed in neuroscience and developmental biology studies, and in the design of controllable cells[93]. The commonly used algal channelrhodopsin ChR2 directly mediates ion transport, bypassing the involvement of G proteins, enzymes, and ion channels necessary for the action of typical animal rhodopsins, making the phototransduction instantaneous[94]. Analogously, the AUREO1c-*LHCX* promoter module circumvents a need for additional E3 ubiquitin ligase proteins and separate transcription factors. The rapid (2–3 min) response time of AUREO1c may therefore expand the light-dependent transcriptional regulatory toolbox and help overcome kinetic challenges in this field[95,96]. More broadly, our study demonstrates the promise of studying organisms occupying diverse niches; environmental fluctuations occur at a wide range of timescales in nature, providing a large, mostly unexplored gene reservoir for synthetic biology uses[97–99].

## Methods
### Algal strains and growth conditions
The wild-type *Phaeodactylum tricornutum* (*P. tricornutum*) strain CCMP2561 was acquired from The National Center for Marine Algae and Microbiota (https://ncma.bigelow.org/). *P. tricornutum* cells were

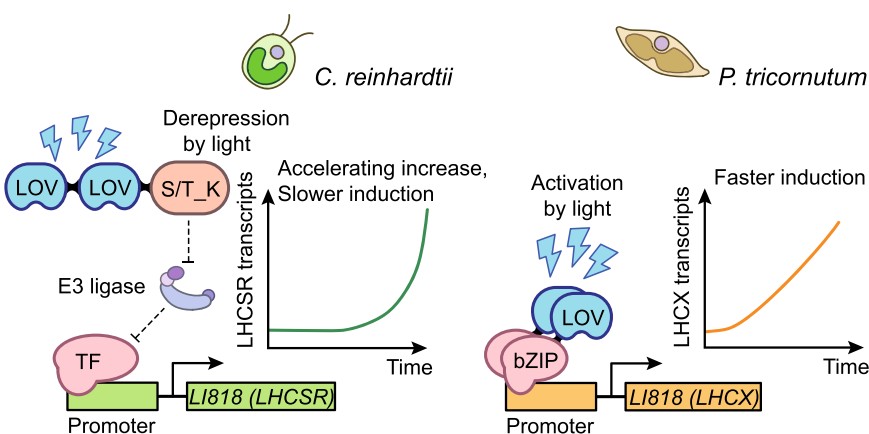

**Fig. 5 | A hypothetical model for the kinetic properties of the pathways regulating photoprotective gene expression in *P. tricornutum* and *C. reinhardtii*.** Note that factors beyond the photoreceptor-mediated pathways may also contribute to the kinetic features of *LI818* gene induction, as cautioned in the "Discussion".

grown in f/2 medium at 21 °C. Solid f/2 medium was supplemented with 1% agar. Liquid cultures are shaken at a speed of 120 rpm and grown under white light (40 µmol photons m$^{-2}$ s$^{-1}$), which is defined as the growth light (GL) condition. Other light conditions used are described in the corresponding figure legends. For RT-qPCR analysis of *LHCX* gene expression on cells shifted from darkness to light, the cells were incubated in darkness for a period of 12 h before the transition. For white light, intensities in the range of 400–550 µmol photons m$^{-2}$ s$^{-1}$ were referred to as "high light (HL)" or "high white light (HWL)" in the "Results"; and intensities of 1200 µmol photons m$^{-2}$ s$^{-1}$ or above were referred to as "very high light (VHL)." The light conditions used for transcriptome analysis are additionally provided in the corresponding "Methods" section below. All the light sources were light-emitting diodes. The wavelengths for blue and red light were 450 ± 5 nm and 660 ± 5 nm respectively. The light spectrum of each culture chamber is shown in Supplementary Fig. 2. The conversion from radiometric units [W/m$^2$/nm] to PAR (µmol photons m$^{-2}$ s$^{-1}$) follows the formula: PAR = $I × λ × 0.836 × 10^{-2}$, where $I$ is the light intensity in W/m$^2$/nm and $λ$ is the wavelength in nm. PUR is calculated from PAR using the formula: $\int_{400}^{700} PAR(λ)A(λ)d(λ)$, where $λ$ denotes the wavelength and $A(λ)$ is the normalized absorption spectrum. Conversion of the photosynthetic active radiation (PAR) to the photosynthetic usable radiation (PUR) was based on the light absorption spectrum of *P. tricornutum* and *C. reinhardtii* cells and light spectrum of each culture chamber. The conversion was according to a formula in published work[100,101]. The light intensity and quality details for each experiment are shown in Supplementary Data 2.

For the green alga *Chlamydomonas reinhardtii*, we employed the widely used strain CC-5325 (aka. CC-4533)[102]. Cells were grown in the Tris-acetate-phosphate (TAP) medium[103] under 22 °C. Both TAP and the acetate-free high-salt (HS) medium[104] have been used for transcriptional studies of *LHCSR* genes[40] and both were employed in our studies. When HS medium was used, cells grown in TAP were washed twice with HS medium, resuspended in HS medium and incubated overnight before our light treatments.

### CRISPR/Cas9-mediated mutagenesis

CRISPR/Cas9 mutagenesis, including guide design[105], construction of vectors expressing Cas9 and guide RNAs[106], and conjugation-based transformation[107], was carried out using the episome-based procedure described in our previous research, which generated the *zep1-b* mutant[108]. Primers that were annealed for generation of the guide RNA template are provided in Supplementary Data 1. The vectors contained a Blasticidin S Deaminase gene expression cassette that allowed for selection of successfully transformed clones on solid f/2 medium containing 5 µg/ml blasticidin S (60218ES50, Yeasen Biotechnology).

The primary transformants underwent PCR amplification and sequencing of the guide region, yielding either completely wild-type or mixed sequencing results; note that we designed primers (Supplementary Data 1) from regions without polymorphisms between the two chromosomes to avoid missing the amplification of one allele. Colonies with mixed sequencing outcomes were then cultured in liquid medium and subsequently spread onto selective solid medium for another round of colony formation. The sub-clones derived from these colonies could present with wild-type sequence, mixed sequencing results, or unequivocal sequencing results. Those sub-clones still yielding mixed results could be either heterozygous or mosaic, containing a mix of mutant and wild-type cells; and the two possibilities cannot be easily distinguished in our procedure. Therefore, sub-clones with unequivocal sequencing results were selected for further physiological analyses. Note that although *P. tricornutum* is a diploid, it exhibits a high frequency of gene conversion[58,74]. This characteristic has made it easy to obtain biallelic (apparently homozygous) mutants for each targeted gene.

### Complementation of the *aureo1c-1* mutant (generation of the COMP line)

The coding sequence of *AUREO1c* was PCR amplified as two fragments from total cDNA using the Phanta Polymerase (Vazyme) and two pairs of primers (the HZ-797/HZ-798 pair and the HZ-799 and HZ-800 pair; Supplementary Data 1). The two fragments were integrated into the homemade pHZ-174 expression vector by homologous recombination (ClonExpress II One Step Cloning Kit, Vazyme). The primers HZ-798 and HZ-799 were designed to introduce synonymous mutations at the sgRNA site to prevent the transgene from being edited by Cas9 expressed from the vector used in initial mutagenesis of *AUREO1c*. The pHZ-174 destination vector contained the endogenous *P. tricornutum* EF1α promoter to drive the expression of the *AUREO1c* gene and a GFP open reading frame to be fused to the C-terminus of AUREO1c. In addition, it contained the *Ble* marker gene that confers resistance to zeocin[109]. The final construct was transformed into *aureo1c-1* cells by electroporation[110]. After 30 days of selection on solid f/2 medium containing 75 µg/ml zeocin (R25001, Thermo), several clones were obtained, with one of them positive in the analysis of GFP fluorescence by flow cytometry (Supplementary Fig. 11). This line was designated the COMP line and stably maintained for further analyses.

### Immunoblotting to verify AUREO1c-GFP expression in the COMP line

A total of $3 × 10^7$ diatom cells were collected by centrifugation and subsequently disrupted by sonication using Bioruptor® Plus (Diagenode). Total proteins were extracted by RIPA buffer (P0013B, Beyotime) with concentrations determined using a bicinchoninic acid

assay kit (C503021-0500, Sangon Biotech) according to the manufacturer's instructions. After being denatured at 95 °C for 5 min and separated on a premade denaturing polyacrylamide gel, proteins were transferred onto a polyvinylidene difluoride membrane for blocking and incubations with antibodies. The primary and secondary antibodies were from SICGEN (AB2166; 1:2000 dilution) and Beyotime (A0181; donkey anti-goat, horseradish peroxidase-conjugated, 1:5000 dilution) respectively. Detection was performed using the Immobilon Western Chemiluminescent HRP Substrate kit (WBKLS0500, Millipore) according to the manual. Wild-type *P. tricornutum* cells were used as a negative control, and blotting using an antibody (AF5012, Beyotime, 1:2000 dilution) against α tubulin was used as the loading control.

## Whole-cell proteomics

Proteomic analyses were performed at the Biomedical Research Core Facilities of Westlake University. *P. tricornutum* or *C. reinhardtii* cells were harvested by centrifugation and lysed by sonication in lysis buffer (RIPA P0013B, Beyotime) containing phosphatase and protease inhibitors (P1045 cocktail, Beyotime). For each strain and condition analyzed, two independent cultures were processed. The same amount of protein extract from each sample replicate was digested using trypsin, targeting cleavage sites at the lysine and arginine residues, and derivatized to harbor tandem mass tags[111] (TMTs) using the TMT10plex™ (for *P. tricornutum*; 90110, Thermo) and TMT6plex™ kits (for *C. reinhardtii*; 90061, Thermo) respectively, following the manufacturer's instructions. The TMT-labeled peptides were separated by a 135 min gradient elution at a flow rate of 0.300 µl/min using a Thermo UltiMate 3000-HPLC system directly interfaced with an Orbitrap Fusion Lumos Tribrid Mass Spectrometer (Thermo) equipped with FAIMS. The analytical column was a home-made fused silica capillary column (75 µm ID, 250 mm length; Upchurch, Oak Harbor, WA) packed with C-18 resin (300 Å, 2 µm, Varian, Lexington). Mobile phase A consisted of 0.1% formic acid in water, and mobile phase B consisted of 80% acetonitrile and 0.1% formic acid. The mass spectrometer was operated in data-dependent acquisition mode using Xcalibur 4.4 software, and the −40 V −55 V, and −70 V CV values of FAIMS were set as the three independent acquisition events. Under one event, a single full-scan mass spectrum in the Orbitrap (375–1800 $m/z$, 120,000 resolution) was followed by several data-dependent MS2 and MS3 scans. The ten most intense ions were first isolated for ion trap CID-MS2 at a precursor ion isolation width of 2 $m/z$, using a maximum ion accumulation time of 35 ms. Directly after each MS2 experiment, the most intense fragment ion in an $m/z$ range between 375–1800 $m/z$ was selected for HCD-MS3. The MS3 AGC was $1 \times 10^5$ and the maximum ion time was 200 ms. Normalized collision energy was set to 35% and 65% at an activation time of 10 ms and 35 ms for MS2 and MS3 scans, respectively. Each mass spectrum was analyzed using the Thermo Xcalibur Qual Browser (4.4) and Proteome Discoverer (2.5.0.400) for the database searching against ASM15095v2_bd (Annotation Source: Ensemble) for *P. tricornutum* and *Chlamydomonas reinhardtii* v5.6 (Annotation Source: JGI Genome Portal) for *C. reinhardtii*, and TMT quantification (Supplementary Data 7–9). The Sequest search parameters included a 10 ppm precursor mass tolerance, 0.02 Da fragment ion tolerance, and up to 2 internal cleavage sites. Fixed modifications included cysteine alkylation and TMT10 modification, and the methionine oxidation was a variable modification. The minimum length of peptides was six and the peptides were filtered at a 1% false discovery rate (FDR). An ANOVA test was employed for comparing individual proteins across different conditions. The protein inventory (Supplementary Data 7–9) encompasses a minimum of one unique peptide for each protein, with the precise tally of unique peptides detailed within the respective datasets. Specifically, proteins of interest such as PtLHCX2, PtLHCX3, and CrLHCSR3 were identified with a minimum of two unique peptides each.

## Chlorophyll fluorescence techniques

For the analysis of the maximal quantum efficiency of photosystem II, approximately $5 \times 10^6$ cells were collected by centrifugation and resuspended in 7 µl of f/2 medium and subsequently spotted onto a 96-well plate with solid f/2 medium. For Supplementary Fig. 8, approximately $5 \times 10^6$ cells were collected and suspended in 200 µl f/2 medium. The cells were dark-adapted for 40 min prior to the measurements of minimal ($F_o$) and maximal chlorophyll fluorescence ($F_m$) by using the Imaging-PAM M-Series platform (Walz, blue light version). The maximal quantum efficiency of photosystem II was calculated as $F_v/F_m = (F_m - F_o)/F_m$[62] and exported from the equipped software as a false-color image.

For measurements of NPQ in *P. tricornutum*, a total of $4.5 \times 10^7$ cells were harvested and resuspended in 2 ml f/2 medium, and dark-adapted for 40 min with shaking before NPQ detection. For *C. reinhardtii*, around $5 \times 10^6$ cells were used and dark-adapted for 15 min. NPQ was measured on the cell suspensions using Dual-PAM-100 (Walz, red light version) as previously described and calculated using the formula NPQ = $(F_m - F_m')/F_m'$[49,112]. $F_m$: maximal fluorescence in the dark-adapted state; $F_m'$: maximal fluorescence at each illuminated timepoint.

For detection of cellular chlorophyll fluorescence after VHL treatment, cells were subjected to flow cytometry (CytoFLEX, Beckman Coulter). The excitation/emission wavelengths are 488 nm/690 nm. The FlowJo software (BD Biosciences) was used for data visualization.

## Pigment analyses by high-performance liquid chromatography (HPLC)

For Supplementary Table 1, cells of wild type, the *aureo1c-1* mutant, and the COMP line were initially grown under 40 µmol photons m$^{-2}$ s$^{-1}$ of white light and transferred to 1200 µmol photons m$^{-2}$ s$^{-1}$ of white light for 6 h and 1 day. For Supplementary Table 2, cells of wild type, the *aureo1c-1* mutant, and the COMP line were initially grown under 40 µmol photons m$^{-2}$ s$^{-1}$ of white light and treated by switching to 900 µmol photons m$^{-2}$ s$^{-1}$ of white light for 10 min or 2 days. Cells were then collected by centrifugation, with pigments extracted in 90% aqueous acetone (v/v) containing 10 mM ammonium acetate, facilitated by sonication. The pigment extracts were analyzed using a UPLC-MS system (Waters/Acquity) equipped with photodiode array (PDA) & QDa detectors. The ACQUITY UPLC HSS T3 C18 column (3 µm, 100 Å, 2.1 × 150 mm, Waters) was used for resolving the pigments. Identities of pigments were assigned based on absorption properties measured by the PDA. Compositions of the mobile phase, flow rate, and the temperature of chromatography were as previously reported[113]. Xanthophyll de-epoxidation state (DES) was calculated as the ratio of diatoxanthin to the sum of diadinoxanthin and diatoxanthin.

## Reverse transcription and quantitative real-time PCR (RT-qPCR)

Cells were collected by centrifugation and subjected to RNA extraction. For the time course experiment (Fig. 4a and Supplementary Fig. 17), at each timepoint, cells were quickly mixed with equal volumes of stop solution (60% ethanol, 2% phenol, 10 mM EDTA; pre-cooled at −20 °C) to quench their transcription[114] before being centrifuged. For other experiments including light treatment shorter than 1 h (see details in Supplementary Data 2), the cells were also mixed with a stop solution in a 1:1 volume ratio. Total RNA was extracted using the RNA Easy Fast Plant Tissue Kit (DP452, Tiangen; for *P. tricornutum*) and the RNeasy Plus Mini Kit (74136, Qiagen; for *C. reinhardtii*) respectively, following the manufacturers' instructions. Reverse transcription was performed using the HiScript III RT SuperMix for qPCR (+gDNA wiper) kit (R323, Vazyme). Each qPCR reaction consisted of 10 µl of ChamQ Universal SYBR qPCR Master Mix (Q711, Vazyme), 8.2 µl of water, 1 µl of cDNA template, and 0.4 µl each of forward and reverse primers (concentration before mixing: 10 µM). RT-qPCR was performed on the qTOWER$^3$G system (Analytik Jena). The *RPS* and *RACK1* genes were used as reference genes for *P. tricornutum* or *C. reinhardtii*

respectively. The delta-delta Ct method[115] was used for relative quantification of transcript abundance of target genes across samples. Primers employed for *PtLHCX2*, *PtLHCX3*, *CrLHCSR3.1*, *CrLHCSR3.2*, and the reference genes are provided in Supplementary Data 1.

## Transcriptome sequencing

Cells of wild type and the *aureo1c-1* mutant were grown in standard growth light (GL) conditions and then transferred to the following conditions: high white light (HWL; 430 μmol photons $m^{-2} s^{-1}$), high red light (HRL; 440 μmol photons $m^{-2} s^{-1}$), high blue light 1 (HBL1; 260 μmol photons $m^{-2} s^{-1}$), and high blue light 2 (HBL2; 430 μmol photons $m^{-2} s^{-1}$) and cultured for 1 h with shaking. The calculation of PUR was performed as described above in the "Algal strains and growth conditions" section. For calculation of the blue light fraction of the PAR, photons with a wavelength within 400–500 nm were included. The results are shown in Fig. 2b and Supplementary Data 2.

For transcriptome sequencing, a total of $1 \times 10^8$ cells were harvested by centrifugation and flash frozen in liquid nitrogen. RNA was extracted with the RNeasy Plus Mini Kit (74136, Qiagen). Library preparation and sequencing on the Nova-PE150 platform (Illumina) was performed by Novogene Co as a service.

## Transcriptome analyses

The raw RNA sequencing (RNA-Seq) reads in the FASTQ format were filtered using fastp[116] (parameters: q, 20; u, 30; n, 5; l, 130) and then mapped to genomic loci[117] using HISAT2[118]. Sequence alignment files in the SAM format were sorted using the SAMtools[119], with read counts exported[120]. For *LHCX2*, reads aligned to Chromosome 1: 2,471,163–2,472,285 (Phatr2_54065) instead of the entire Phatr3_EG02404 gene model (Chromosome 1: 2,469,549–2,472,285) were included; the Phatr2_54065 model is supported by our RNA-Seq reads and PCRs (Supplementary Fig. 16). Read count normalization (Supplementary Data 4) and principal component analysis (PCA) were performed using the DESeq2 environment[121].

To compare the transcript abundances between wild type and the *aureo1c-1* mutant for individual genes, the *p* value and the false discovery rates (FDRs) were calculated using the *t*-test and the Benjamini–Hochberg multiple testing procedure[122] respectively (Supplementary Data 5). Prism 8.0 (Graphpad) was used to plot the correlations between the log$_2$(WT/*aureo1c-1*) values under different light conditions and yield the Pearson's correlation values (Fig. 2d).

To distinguish between AUREO1c-dependent and -independent genes among those regulated by high light in wild type (Fig. 3a), we defined a gene as AUREO1c-dependent if it satisfied one of the two criteria below regarding the ratio of the normalized read counts: (1) (WT-HWL)/(WT-GL) > 4 and (WT-HWL)/(*aureo1c-1*-HWL) > 2; (2) (WT-HWL)/(WT-GL) < 0.25 and (WT-HWL)/(*aureo1c-1*-HWL) < 0.5.

A gene was defined AUREO1c-independent if it satisfied one of these two criteria: (1) (WT-HWL)/(WT-GL) > 4 and (WT-HWL)/(*aureo1c-1*-HWL) < 1.6; (2) (WT-HWL)/(WT-GL) < 0.25 and (WT-HWL)/(*aureo1c-1*-HWL) > 0.625.

Enrichments of gene ontology (GO) terms among regulated genes were calculated using the software clusterProfiler[123] (v4.4.4), which employs the Benjamini–Hochberg multiple testing procedure[122] to derive adjusted *p* values (Supplementary Data 6).

## Fluorescence microscopy to determine AUREO1c localization

Cells of the COMP lines or wild-type cells grown under normal light (40 μmol photons $m^{-2} s^{-1}$ of white light) or high light (550 μmol photons $m^{-2} s^{-1}$ of white light) were mounted with equal volumes of SlowFade Glass Soft-set Antifade Mountant (S36920, Thermo) containing 4′,6-diamidino-2-phenylindole (DAPI) and imaged on a live-cell imaging microscope (DeltaVision, GE) with an oil immersion objective (Olympus 100 × /1.40). Excitation/emission collection wavelengths used for chlorophyll autofluorescence, DAPI, and GFP

were: 632 nm/679 nm, 390 nm/435 nm, and 475 nm/525 nm respectively. For each channel, the same exposure settings were applied onto cells of wild type and cells of the COMP line.

## Production of recombinant His-tagged or glutathione S-transferase (GST)-tagged AUREO1c protein for in vitro experiments

To obtain His-tagged AUREO1c proteins, the cDNA sequence of AUREO1c was PCR amplified from total cDNA using primers HZ-1043 and HZ-1044 (Supplementary Data 1) and cloned into the pET28a vector (69864, Novagen) by homologous recombination using the ClonExpress II One Step Cloning Kit (C112, Vazyme); the vector contained a 6x-His tag at the N-terminus of the integrated gene. The completed construct was transformed into Transetta (DE3) *E. coli* strain (CD801, TransGen). Expression of AUREO1c was induced by supplementation with isopropyl ß-D-1-thiogalactopyranoside (IPTG) to a final concentration of 0.1 mM. Cells were broken by a high-pressure homogenizer (AH-Basic, ATS) and the recombinant AUREO1c proteins were purified by affinity chromatography on a gravity column using a Ni-NTA His binding resin (88222, Thermo). A further round of purification was achieved by fast protein liquid chromatography (ÄKTA pure™ 25, GE) using the HiTrap Q HP column (Cytiva). Finally, the proteins were concentrated by ultrafiltration using Amicon MWCO 30 kDa Ultra tubes (UFC903008, Millipore). Protein concentrations were determined using the Bradford method[124].

To obtain GST-tagged AUREO1c proteins, the cDNA sequence of AUREO1c was cloned using primers HZ-1387 and HZ-1388 into pGEX4T-1, also by homologous recombination. Induction of protein expression, cell lysis, and protein concentration measurements were performed as described above for His-tagged AUREO1c proteins. Purification of GST-AUREO1c was conducted using the GSTSep Glutathione MagBeads (20562ES08, Yeasen). Unfused GST was purified to be used as a negative control in experiments afterwards.

## In vitro analyses of AUREO1c responses to blue light

Recombinant His-tagged AUREO1c proteins were diluted by using 1 × phosphate-buffered saline (PBS) to a concentration of 5 μM and incubated in the dark or under blue light (200 μmol photons $m^{-2} s^{-1}$) for 60 min. They were then crosslinked by the supplementation of bis(sulfosuccinimidyl)suberate (BS$^3$) to a final concentration of 5 μM. After 60 min of incubation on ice, the crosslinking was quenched by the addition of Tris to 50 mM. Incubation continued for 15 min before further analyses.

One fraction of the samples was loaded onto the ÄKTA pure™ 25 platform for size-exclusion chromatography (SEC; Fig. 2e), using the Superdex 200 Increase 10/300 GL column. Absorbance at 280 nm was recorded.

Another fraction was subjected to ultra high-performance liquid chromatography-native mass spectrometry (UHPLC−nMS) analysis (Supplementary Fig. 15) on the Thermo Vanquish UHPLC system coupled to a Thermo Q Exactive UHMR mass spectrometer. The LC separation was carried out on a Waters BEH SEC column (2.1 × 150 mm, 1.7 μm) at room temperature, and isocratic separation was implemented using 100 mM ammonium acetate as the mobile phase with the flow rate at 0.2 ml/min. The MS parameters used were as follows: capillary voltage, 3500 V in positive mode; resolution, 12,500; SID voltage, 50 eV; capillary temperature, 275 °C; sheath gas, 20; aux gas, 5. Full scan spectra were acquired with a mass range from 1000 to 8000 *m/z*, and the molecular weight range of the protein was calculated manually.

For measurements of absorption properties, recombinant His-tagged AUREO1c proteins were first purified by Ni$^{2+}$-column affinity purification and then further purified using the Superdex 200 column on the ÄKTA pure™ 25 system. The fraction containing AUREO1c was diluted by using 1 × HEPES buffer (200 mM NaCl contained, pH: 7.9) to

a concentration of 5 μM and incubated in darkness or under blue light (1000 μmol photons m$^{-2}$ s$^{-1}$) for 15 min. Then absorption spectra were measured on a microplate reader (Thermo Varioskan LUX).

## DNA affinity purification sequencing (DAP-Seq)

DAP-seq and sequencing read alignment were performed as described in a published protocol[125], with minor modifications. Genomic DNA from wild-type cells was fragmented using a sonifier (ME220, Covaris), with the following parameters: "duty factor", 15%; "peak power", 50; "cycles/burst", 200; "duration": 200 s. The end-repair reactions, A-tailing reactions, and adapter ligation reactions were carried out using the Fast DNA End Repair kit (K0771, Thermo), the DNA Polymerase I Klenow Fragment exo- kit (N105, Vazyme), and the T4 DNA Ligase kit (M1804, Promega), respectively, according to the manufacturer's instructions. The adapter for ligation was generated by annealing of primers HZ-1795 and HZ-1796. After ligation, the mixture was purified using the GeneJET PCR Purification Kit (K0701, Thermo). For each DNA affinity purification reaction, 500 ng of ligation products were incubated with 5 μg of GST-AUREO1c proteins or GST proteins under room temperature. The incubation lasted for 1 h. The beads were washed 6 times with PBS supplemented with IGEPAL CA-630 (I8896, Sigma; final concentration: 0.005%) and additionally washed twice with PBS. The eluted DNA was used to construct the sequencing library by PCR using primers HZ-1797/HZ-1800 (for GST) or HZ-1799/HZ-1800 (for GST-AUREO1c), followed by gel purification using the E.Z.N.A. Gel Extraction Kit (D2500, Omega). Sequencing of the libraries were performed by Novogene Co on the Illumina Nova-PE150 platform. The reads are illustrated using the Integrative Genomics Viewer (IGV) software[126] (Fig. 3e).

## Electrophoresis mobility shift assay (EMSA)

Promoter sequences of *LHCX2*, *LHCX3*, and the alpha tubulin gene were PCR amplified from genomic DNA of *P. tricornutum*, with one of the primers biotinylated at the 5' terminus (Supplementary Data 1), and then purified. Incubation of recombinant His-tagged AUREO1c with the probes, mobility analysis of probes, and their visualization was performed using the Light Shift Chemiluminescent EMSA Kit (20148, Thermo) according to the manufacturer's instructions. Control samples that did not contain AUREO1c proteins were included to show the mobility of free probes. In additional samples, non-biotinylated probes were added to compete for binding by AUREO1c.

## Accession numbers

For *P. tricornutum* genes, Phatr3 IDs of the ASM15095v2 genome assembly are provided as follows. *AUREO1a*: Phatr3_J8113; *AUREO1b*: Phatr3_J15977; *AUREO1c*: Phatr3_J51933; *LHCX2*: Phatr3_EG02404 (Phatr2_54065); *LHCX3*: Phatr3_J44733; *RPS*: Phatr3_J10847; *EF1α*: Phatr3_J18475; α tubulin: Phatr3_J54534. For *C. reinhardtii* genes, v5.6 IDs of the ABCN02000000 genome assembly are provided as follows. *LHCSR3.1*: Cre08.g367500; *LHCSR3.2*: Cre08.g367400; *RACK1*, Cre06.g278222.

Raw transcriptome files from the sequencing platform have been uploaded to NCBI Sequence Read Archive (SRA; https://www.ncbi.nlm.nih.gov/sra). Accession numbers for each individual transcriptome are provided in Supplementary Data 3. No custom code was used to derive major conclusions in this study.

## Reporting summary

Further information on research design is available in the Nature Portfolio Reporting Summary linked to this article.

## Data availability

All generated and analyzed data from this study are included in the main figures, Supplementary Information, or have been uploaded to public repositories. All unique materials are readily available from the corresponding author on request. RNA sequencing data generated in this study have been deposited in the NCBI database under accession code. The accession number of sequence Read Archive (SRA) data are SAMN34393917, SAMN34393918, SAMN34393919 (for WT in growth light), SAMN34393932, SAMN34393933, SAMN34393934 (for *aureo1c-1* in growth light), SAMN34393911, SAMN34393912, SAMN34393913 (for WT in high red light), SAMN34393926, SAMN34393927, SAMN34393928 (for *aureo1c-1* in high red light), SAMN34393914, SAMN34393915, SAMN34393916 (for WT in high white light), SAMN34393929, SAMN34393930, SAMN34393931 (for *aureo1c-1* in high white light), SAMN34393905, SAMN34393906, SAMN34393907 (for WT in high blue light 1), SAMN34393920, SAMN34393921, SAMN34393922 (for *aureo1c-1* in high blue light 1), SAMN34393908, SAMN34393909, SAMN34393910 (for WT in high blue light 2), SAMN34393923, SAMN34393924, SAMN34393925 (for *aureo1c-1* in high blue light 2). Accession numbers for each individual transcriptome are also provided in Supplementary Data 3. The mass spectrometry proteomics data have been deposited to the ProteomeXchange Consortium via the PRIDE[127] partner repository with the dataset identifier PXD045342 for *Phaeodactylum tricornutum*, PXD045344 for *Chlamydomonas reinhardtii* carried out in TAP medium, and PXD050435 for *Chlamydomonas reinhardtii* carried out in HS medium. The DAP-seq data used in this study are available in the NCBI database under accession code SAMN37389769 (incubating AUREO1c-GST fusion protein with DNA in vitro) and SAMN37389768 (incubating GST protein with DNA in vitro, control). Source data are provided with this paper.

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

## Acknowledgements

The authors thank these colleagues for helpful discussion: Hongtao Liu from Shenzhen University; Congming Lu from Shandong Agricultural University; Yizhong Yuan from the Carnegie Institution for Science; Deqiang Duanmu and Longlong Wang from Huazhong Agricultural University; Wenbin Zhou, Zefu Lu, Pan Liu, and Shaobo Wei from the Chinese Academy of Agricultural Sciences; Qiguang Xie from Henan University; Xiaojie Pang from Chinese Academy of Sciences; Mingzhu Fan, Yanlei Feng, Xiangdong Fu, Yinping Gao, Peng Liu, Fang Xiao, Yanxiao Zhang, and Jia Zheng from Westlake University. We thank Du Cao (Westlake University) for proofreading the manuscript. We also acknowledge the Instrumentation and Service Center for Molecular Sciences and the Biomedical Research Core Facilities at Westlake University for technical support. This work was supported by fundings listed below: National Key R&D Program of China (2019YFA0906300 to X.L.), National Natural Science Foundation of China (32150022 to X.L., 42106114 to H.Z., 42206115 to T.C.), National Key R&D Program of China (2022YFC3401800 to T.C.), Zhejiang Province Key R&D Program (2023SDXHDX0002 to X.L.), Zhejiang Natural Science Foundation (LR20C020002 to X.L.), Zhejiang Provincial Key Laboratory Construction Project (to X.L.), Westlake Research Center for Industries of the Future (WU2023C002 to X.L.), Westlake Education Foundation (WU2023B002 to X.L.), and the Westlake Center for Genome Editing (to X.L.).

## Author contributions

H.Z. and X.L. conceived the project. H.Z., L.T., and X.L. designed the research. H.Z., K.G., X.X., M.Z., T.C., Y.Y., J.S., J.C., J.Z., Y.J., and S.F. performed experiments and bioinformatic analyses; X.L. wrote the manuscript with contributions from all authors.

## Competing interests

X.L., H.Z., K.G., X.X., and T.C. have submitted a Chinese patent application for this work presented in the manuscript. The application number is 202410675502.8. The other authors declare that they have no competing interests.
