## [Peer Review File · Nature Communications]

A rapid aureochrome opto-switch enables diatom acclimation to dynamic lightEditorial note: Parts of this Peer Review File have been redacted as indicated to remove third-party material where no permission to publish could be obtained.

REVIEWER COMMENTS

Reviewer #1 (Remarks to the Author):

Light is the driving force for photosynthesis but all photosynthetic organisms are often exposed to light intensities that exceed their photosynthetic capacity, potentially leading to photodamage, unless efficiently dissipated as heat via the photoprotective mechanism of non-photochemical quenching (NPQ). Blue light perception plays a dominant role in the regulation of photoprotection; in Arabidopsis the blue light photoreceptors phototropins (PHOTs) control chloroplast positioning, a mechanism that minimizes exposure of chloroplasts to light under strong illumination; in green algae (*Chlamydomonas reinhardtii*), PHOT transcriptionally controls LHCSR3, the main protein effector of photoprotection in this alga, and in diatoms (*Phaeodactylum tricornutum*) AUREO1a and AUREO1b have been suggested to be involved in the control of NPQ.

In this manuscript Zhang and colleagues report on their findings linking the blue-light photoreceptor AUREO1c to the transcriptional activation of photoprotection in the marine diatom *Phaeodactylum tricornutum*. They first used CRISP-CAS9 to generate single and double mutants of LHCSX2 and 3 to reveal their overlapping roles in photoprotection in *Phaeodactylum*; they then identified AUREO1c as the blue light receptor mediating LHCX2/3 expression by binding to the DNA of their promoter regions; finally, they compared the kinetics of photoprotective gene mRNA accumulation in *Phaeodactylum* and in *Chlamydomonas*, which makes use of a similar (LOV-based) but different (kinase signalling based) mechanism. The findings of this manuscript are very interesting, the manuscript is very well written and I believe it will attract attention of a wide readership of the journal interested in microalgae physiology, photosynthesis, ecology, signalling, evolution. I have a few specific comments and suggestions.

Fig. 1f provides only a qualitative assessment of photodamage. The authors should indicate

how much chlorophyll accumulates in the strains under the different light conditions.

Supplementary Fig. 7: is the flavin cofactor embedded in the recombinant AUREO1c protein? What are the functional implications of the finding that AUREO1c is dimerized in both light and dark conditions?

What is the role of light intensity in the AUREO1c-dependent gene expression of LHCX2/3? Is the AUREO1c mediated activation of LHCX2/3 an on/off type of response? Are other factors (e.g. redox or ROS signals from the chloroplast) come to play? All data presented here compare normal light to high light conditions, conditions in which AUREO1c is active. What happens when dark-acclimated cells are shifted to low, normal or high light? Would the expression levels of LHCX2/3 be similar across all light intensities?

Fig. 4: The authors set out to compare the receptor-to-nucleus signal transmission that controls photoprotective responses in *Chlamydomonas* and *Phaeodactylum*. I find the idea very interesting, but in my opinion the experimental setup would benefit from some improvements, namely that the same trophic mode should be used in both *Chlamydomonas* and *Phaeodactylum* experiments, since acetate that was present in *Chlamydomonas* experiments, is a suppressor of LHCSR3. Even though LHCSR3 mRNA accumulation does occur even in the presence of acetate, when transitioning from low to high light, the levels are 10-20 times lower as compared to phototrophic conditions. As a result, mRNA quantification in these conditions of low transcript abundance, may suffer from insufficient resolution to properly address the impact of HL in the very first minutes of the exposure. I strongly recommend repeating the *Chlamydomonas* experiment using phototrophic medium. The authors may also want to consider starting from darkness instead from NL, as I also mentioned in my previous comment.

A final comment on the ecological relevance of the results presented in Fig. 1c, f showing that the aureo1c cultures collapse after 2 days at VHL. Is there an observable phenotype (e.g. pigment composition, growth, NPQ) on a time scale of hours, that would be closer to the conditions the cells would experience at their natural habitat?

Minor comment

Fig. 1d: add error bars, p-values

Fig. 1d: Without wanting to deviate the focus of the work from AUREO1c and LHCX2/3, I believe that this manuscript provides an opportunity to get an overview of the roles of the three AUREO isoforms (1a, 1b and 1c) in regulating expression of all four photoprotection genes (LHCX1-4). In this regard I would encourage the authors to add a supplementary Figure with this information.

Reviewer #2 (Remarks to the Author):

A rapid transcriptional opto-switch enables diatom acclimation to dynamic stress.

Zhang et al. use CRISPR/Cas9 methods to knock-out activities of AUREO1a,b and c and LHCX2 and LHCX3 to resolve their interplay and function in acclimation to light stress in the diatom *Phaeodactylum tricornutum*. Using conjugation three mutant lines each of AUREO1a, AUREO1b and AUREO1c were produced and tested under defined light intensities and quality conditions. At 550 $\mu\text{mol photons m}^{-2} \text{s}^{-1}$ maximum quantum yields of PSII was reduced for the *aureo1c* mutant and not in *aureo1a* and *1b* pointing to a role in light acclimation. The *aureo1c* showed strongly reduced survival at light intensities of 1200 $\mu\text{mol photons m}^{-2} \text{s}^{-1}$, and the phenotype of *aureo1c-1* line was rescued by complementation with a *aureo1c*-GFP fusion product. Images (stamp size) indicate that AUREO1c is localized to the nucleus supporting the role as a bZIP/LOV light regulated transcription factor. Transcriptomics of wt, *aureo* mutants and a complemented *aureo1c* mutant showed reduced expression of two light harvesting protein encoding genes LHCX2 and LHCX3 in the *aureo1c* mutant indicated that they are under control of *aureo1c*. LHCX proteins has previously been shown to have a role in NPQ and NPQ was reduced in the *aureo1c* compared to wt at 900 $\mu\text{mol photons m}^{-2} \text{s}^{-1}$.

The data presented point to a model where AUREO1c receive a blue light induced signal activating expression of the LHCX2 and LHCX3 genes which in an not explained mechanism affect NPQ.

The results are very well documented.

Some comments:

Title could include AUREOCHROME opto switch ...

It is previously established that AUREOCHROMES blue light receptors can act as rapid opto switches and that LHCX's are involved in NPQ and acclimation to light conditions. And that they are nucleus localized sensors for high blue light intensities.

The present manuscript provide a further step and a functional analysis of three specific AUREOCHROME's and two LHCX's. Produced results are solid and provide evidence showing that AUREOCHROME1c specifically control LHCX2 and LHCX3 expression levels thereby affecting NPQ in acclimation to (blue)light stress. Protein size exclusion indicate that AUREOCHROME's as many other TFs primarily are dimerized and have some mass variation pointing to modifications at protein level. Complementation of aureo1c support the results.

The results are well presented and the text read well and is easy to follow, with a few exceptions:

Major point:

The documentation of the biallelic mutants of aureo1a, aureo1b and aureo1c should be improved or at least explained better. It is referred to supplementary fig.3 as proof of bi-allelic changes for all 9 mutants of AUROCHROME's and for the 6 mutants of LHCX2 and 3. Only one mutation is specified for each mutated line indicated that all are mono-biallelic? Gene conversion might explain such variants sometimes, but not for five genes at different chromosomal location?

One possibility to overcome this is to search for polymorphisms. These genes are relatively small and have polymorphisms that potentially might clarify if it really is bi-allelic mutants.

Minor points:

Suppl. Fig 3:

Indicate what boxes of the gene models represents at a-e.

The plus/minus strand/orientation of the target gene sequences would be easier to read if it

is presented in an uniform style.

Abstract line32: ...strong blue light in elution profile.. explain?

Li818 vs LHCX. Keep LHCX nomenclature throughout?

P5 line 105: .. cell survival... Likely true, but no staining/cell numbers?

Why was line aureo1c-1 chosen?

Reviewer #3 (Remarks to the Author):

The manuscript deals with the demonstration of the role of a specific blue-light photoreceptor, the aureochrome, in the light response of the model diatom microalga *Phaeodactylum tricornutum*. Using a diversity of approaches and technics, the authors successively show how AUREO1 c is crucial for photoprotective LHCx genes expression and proteins and synthesis, NPQ induction and photosynthesis performance under high light and fluctuating light. They additionally show the nucleus localization of AUREO1 c and how it directly interacts with the promoters of LHCx genes to regulate their expression. By comparing the light-response dynamics of *P. tricornutum* to *Chlamydomonas* in this framework, they finally propose an explanation for the fast induction of photoprotection in diatoms, and how this could be crucial for their ecological success in mixed coastal waters. The data and conclusions are of high importance for understanding the ecological success of diatoms, which is the most diverse and productive eukaryotic group of microorganisms, in modern oceans. Beyond, it brings original data on the role of photoreceptors in photosynthetic organisms, and essential differences among phylogenetic lineages (the green which dominates the land versus the red which dominates the oceans). In general, this piece of work is remarkable by the number of technics used and the amount of data. It is clearly written, well presented, especially the Results section follows a logic step-by-step strategy that is particularly pleasant to read. The figures are also nicely organized and well synthesized. It clearly deserves publication in Nature Communications after some revisions. Here are some general and more specific comments to help the authors improve their

manuscript.

Light:

-it is essential to show the light spectrum of all the light sources used in this work: the white growth light, all types of experimental lights (white, blue, etc.), and PAM-fluorometer, which I guess, used blue light; a good example is Lines 188-189: 'note that strong blue light is a component of HWL', the authors need to show it and to calculate proportions so that the BL component of WL can be directly compared with HBL.

-the PAR versus PUR equivalences between all light sources (growth and experimental) need to be calculated; for instance lines 177-184 and the use of two HBL1 and HBL2 intensities (and corresponding Methods section), this is essential and it needs to be clarified with real numbers in order to avoid foggy arguments such as 'more comparable' (line 183).

-experimental lights: what is the most difficult to follow are the different light treatments that have been used, especially the intensities and durations of light exposures are never the same from an experiment to another, this is not impeding the conclusions per se but that would be nice if the authors could better present (directly on figures ? in a summary Table ?) these differences.

-instead of using NL-normal light (which means nothing in a photophysiological framework), better use GL for growth light.

-as stated by the authors all previous works always used low to moderate intensities of blue light when the current study uses 'high' blue light: it would have been ideal for the present work to additionally include data generated under low blue light to get some kind of comparative baseline with the 'high' blue light. If the authors have such data they should include them, if not, they should, in the Discussion, use previous published works (Kroth et al, and so on) to compare with their data.

Abstract:

-Line 20: Diatoms OFTEN outnumber.

-Line 22: 'are largely elusive', no, this needs to be modulated, there are other places in the text (see comments below) where the authors tend to underestimate previous works, and such statements need to be avoided. Many previous works regarding 'the mechanisms underlying acclimation of diatoms to high light stress' are not elusive at all: they are clear

and strong, and for some of them they simply did not go as far as the present work.

-Line 29, 32 and 34: precise these are named LHCx in diatoms.

-Line 34: expression of SOME LI818 genes: the present work demonstrates such fact only for 2 over 4 LHCx in *Phaeodactylum* (and there are up to 17 LHCx isoforms in other diatom and neighbour haptophyte species).

-Line 39: dynamic LIGHT conditions.

Introduction:

-Line 51: there are more recent references than number 4.

-Line 54: references 5-7, please do not forget recent and important works and reviews by Falciatore et al.

-Line 55: 'translational potential on food crops', this statement is unclear or incomplete and needs to be better explained.

-Line 59: references 8,9: these are really old works and since then some new concepts, especially as regards to photoinhibition, were proposed that are included in more recent works.

-Line 69: works by Lepetit et al. need to be cited here.

-Line 82: 'diatom survival', this is overstated, please modulate; it is not about their survival but about the maintenance of their photosynthetic performance and productivity.

-Line 83: 'await to be demonstrated', this is also overstated, please modulate; there are several recent works showing how LHCx are essential in maintaining photosynthetic performance in diatoms, many of them are cited here at other places in the text (Lepetit, Kroth, Falciatore, Ruban, etc.), and I propose the authors to add this one:

<https://doi.org/10.3389/fpls.2022.841058>

-Line 83 : 'the mechanism by which specific LHCx isoforms are regulated', please precise you refer to their synthesis.

Results:

-Line 105: 'survival', same remark as above, overstated, please change with what you have effectively measured and show (Fv/Fm), which is 'photosynthetic performance'.

-Line 118: 'lower Fv/Fm can arise', to be modulated in 'lower Fv/Fm could arise' or equivalent; a lower Fv/Fm can be due to many quenching events and when the light

intensity is excessive, it is often a mix of qE and qI , qI being itself ill-defined; the only way to conclude on a direct relationship between lower Fv/Fm and PSII photodamage is to proceed with a low light recovery period after the light stress period: the fraction of Fv/Fm that did not recover is equivalent to the fraction of damaged PSII.

-Line 132: 'proteins showed fast recovery after blue light excitation', this statement is unclear and needs better explanation.

-Line 133: 'suggesting', also unclear how the 1st part of the sentence suggests the second part, the whole sentence needs to be re-written.

-Lines 146-150: showing DES data is not enough, the authors need to show 'real' DD and DT pigments concentrations per Chl a and per cell.

-Line 157: but DES is lower in the COMP line (supplementary Fig 5b), the authors need to explain this intriguing result.

-Line 318: 'light fluctuations often last 1-2 h in total', this statement is puzzling to say the less, light fluctuations, especially in coastal systems, are permanent and show different frequencies depending on the scale, this needs to be better explained and justified.

-Part starting Lines 283-284: the FL treatment used here is raw, next time the authors should better use a more refined set-up as the one used in Lepetit et al (ref 32 here) to study the fine dynamics of LHCx synthesis; in general, over this part of the manuscript, former works, such as ref 32, should be used by the authors to compare with their findings, that will likely strengthen their data/conclusions.

-Lines 324-325: 'for sufficient photoprotection', please show the NPQ data to support such statement.

Discussion:

The discussion is a bit strange, it is quite short (due to the fact some specific points have been justified/discussed in the Results section) and it reads more like an extended Conclusion. I am personally fine with it as the Results section reads well, but the Editor might feel different about it.

-Lines 354-360: please cite the Figure 5 already here.

Figures:

-Supplementary figure 1: the localization of green microalgae is unclear, and should be

better shown: are they present in some kind of sea-water cuvettes or is this a representation of a freshwater system ? The boarder between the oceans and the land with their continental aquatic systems (lakes, ponds, rivers, etc.) should be better drawn. Diatom side: the food-web should be better shown too, it is strange that a diatom cell is bigger than a fish. In general, and even if this is a schematic sketch, diatoms and green microalgae should be sown with a smaller size.

We are extremely thankful for the favorable evaluation of our manuscript and the invaluable feedback provided by the Reviewers. Below, we present an overview of the significant revisions that we have implemented in response to the reviewers' comments:

1. Our original presentation may have led readers to perceive AUREO1c as an ON/OFF switch activated by a specific light intensity threshold. We now clarify that (1) the signaling capacities of photoreceptors generally increase synchronously with rising light intensity, and (2) photoreceptors exhibit varying light sensitivities. Related to the first point, we now include data on AUREO1c-dependent *LHCX2/3* expression to illustrate that the function of AUREO1c escalates in response to increasing light intensity. In relation to the second point, we discuss that in plants, both phototropins govern phototropism, yet only the less sensitive one (PHOT2) is utilized to signal high light stress and trigger protection (chloroplast avoidance). In *P. tricornutum*, AUREO1c was shown to be lower than AUREO1a in light sensitivity and this motivated us to investigate its function in photoprotection. Please see responses (4), (24), and (40).

2. Related to the above point, we have performed experiments in the additional light and trophic conditions as suggested by Reviewers. These results do not alter our conclusions but have helped us to further comprehend the photoacclimation of the two algae. Please see responses (4), (6), (7), and (24).

3. In terms of genetics, we now clarify that both homozygous and heterozygous mutants could be obtained. We chose the homozygous ones to characterize because heterozygosity cannot be easily distinguished from mosaicism of the (sub)clone. Please see response (12).

4. As requested by all three reviewers, we have performed a more quantitative investigation of the physiological parameters of *aureo1c* culture under (very) high light conditions, including pigment composition, F_v/F_m , NPQ, cell growth, and chlorophyll fluorescence intensity. Please see responses (1), (8), (18), and (38).

5. We have improved the writing by following the reviewers' suggestions. This includes (1) providing more accurate descriptions on the light conditions used, (2) providing more consistent gene nomenclature and labels for figure elements, (3) discussing the functional implications of our experimental data, (4) citing additional literature as needed, and (5) expanding the discussion to accommodate the additional data.

Below are our point-by-point responses (in blue) to Reviewers' comments (in black). Please refer to the "**Manuscript file with revision marked**" PDF File for the page and line number citations.

Responding to

REVIEWER COMMENTS

Reviewer #1

Light is the driving force for photosynthesis but all photosynthetic organisms are often exposed to light intensities that exceed their photosynthetic capacity, potentially leading to photodamage, unless efficiently dissipated as heat via the photoprotective mechanism of non-photochemical quenching (NPQ). Blue light perception plays a dominant role in the regulation of photoprotection; in *Arabidopsis* the blue light photoreceptors phototropins (PHOTs) control chloroplast positioning, a mechanism that minimizes exposure of chloroplasts to light under strong illumination; in green algae (*Chlamydomonas reinhardtii*), PHOT transcriptionally controls LHCSR3, the main protein effector of photoprotection in this alga, and in diatoms (*Phaeodactylum tricornutum*) AUREO1a and AUREO1b have been suggested to be involved in the control of NPQ.

In this manuscript Zhang and colleagues report on their findings linking the blue-light photoreceptor AUREO1c to the transcriptional activation of photoprotection in the marine diatom *Phaeodactylum tricornutum*. They first used CRISP-CAS9 to generate single and double mutants of LHCSX2 and 3 to reveal their overlapping roles in photoprotection in *Phaeodactylum*; they then identified AUREO1c as the blue light receptor mediating LHCX2/3 expression by binding to the DNA of their promoter regions; finally, they compared the kinetics of photoprotective gene mRNA accumulation in *Phaeodactylum* and in *Chlamydomonas*, which makes use of a similar (LOV-based) but different (kinase signalling based) mechanism. The findings of this manuscript are very interesting, the manuscript is very well written and I believe it will attract attention of a wide readership of the journal interested in microalgae physiology, photosynthesis, ecology, signalling, evolution. I have a few specific comments and suggestions.

We greatly appreciate your recognition of our work and have subsequently conducted a series of experiments based on your suggestion.

Fig. 1f provides only a qualitative assessment of photodamage. The authors should indicate how much chlorophyll accumulates in the strains under the different light conditions.

(1) Thanks for your suggestion. We adopted your suggestion to quantify chlorophylls and normalized their content based on cell number (newly added **Supplementary Table 1**). We found that the *aureo1c-1* mutant had a similar pigment composition with wild type under normal light (growth light, GL). After six hours of VHL treatment, the mutant was still similar to wild type and neither showed a major decrease in chlorophyll content from GL. And we chose this time point and earlier time points for physiology measurements, as shown in response (8), because analyses on bleached cells would not be meaningful. After 1 d of treatment, wild type decreased chlorophyll *a* content by approximately 8 fold from GL, whereas the mutant was 7 times lower than the wild-type control, showing a 60-fold decrease from GL. We could see a difference by naked eyes at this time point. After 2 d of treatment, the culture of the mutant was severely bleached (shown in **Fig. 1b, f**). In our HPLC, the pigment peaks were barely detectable and could not be reliably quantified from the mutant samples. Thus we included GL, 6 h VHL, 1 d VHL samples in **Supplementary Table 1**. In addition to chlorophyll quantification by HPLC, we also demonstrated the loss of chlorophylls in the mutant using a different means: most of the mutant cells lost chlorophyll fluorescence after 1 d of VHL treatment as shown by flow cytometry (newly added **Supplementary Fig. 7b**). These new results corroborate that AUREO1c is indeed for acclimation to very high light stress. We have incorporated and discussed these results (lines 225-233, 245-250).

Samples	GL			VHL, 6 h			VHL, 1 d		
	WT	aureo1c-1	COMP	WT	aureo1c-1	COMP	WT	aureo1c-1	COMP
Fx (fg/cell)	789 ± 25	943 ± 33	868 ± 35	699 ± 46	761 ± 9	740 ± 18	224 ± 8	24 ± 8	239 ± 9
Chl c ₁ (fg/cell)	106.0 ± 4.6	123.1 ± 3.7	117.4 ± 5.3	103.0 ± 6.2	113.9 ± 3.7	107.7 ± 9.1	35.7 ± 1.2	3.4 ± 2.4	38.7 ± 3.5
Chl c ₂ (fg/cell)	110.9 ± 4.1	141.0 ± 5.3	121.5 ± 5.4	96.6 ± 4.9	109.8 ± 2.2	100.7 ± 1.2	23.3 ± 1.0	1.8 ± 1.1	26.7 ± 2.3
Ddx (fg/cell)	123.3 ± 10.8	127.8 ± 2.5	128.5 ± 6.3	64.4 ± 3.0	20.9 ± 3.3	93.4 ± 8.2	16.6 ± 0.7	1.4 ± 1.1	53.8 ± 4.4
Dtx (fg/cell)	3.8 ± 0.4	4.4 ± 0.4	3.8 ± 0.2	103.0 ± 14.3	137.9 ± 7.7	79.0 ± 5.7	52.3 ± 3.2	1.9 ± 1.1	47.1 ± 4.4
Chl a (fg/cell)	806 ± 25	938 ± 33	882 ± 35	755 ± 46	821 ± 9	783 ± 18	105 ± 8	15 ± 9	103 ± 9
β-car (fg/cell)	58.0 ± 1.3	63.4 ± 2.7	60.2 ± 2.0	57.6 ± 2.1	65.8 ± 1.2	52.3 ± 2.8	3.4 ± 0.3	n.d.	4.7 ± 1.1
DES	0.031 ± 0.005	0.035 ± 0.003	0.030 ± 0.001	0.620 ± 0.042	0.872 ± 0.013	0.465 ± 0.020	0.764 ± 0.004	0.592 ± 0.113	0.474 ± 0.025

newly added **Supplementary Table 1**

Panel b of the newly added **Supplementary Fig. 7**

Supplementary Fig. 7: is the flavin cofactor embedded in the recombinant AUREO1c protein?

(2) This is an important question because the cofactor is essential for aureochromes to undergo a dark-light transition response. The flavin cofactor confers an absorbance peak around 450 nm. It is available from *E. coli* and our recombinant proteins appeared yellow by naked eyes. We have now measured the absorption spectrum of the proteins incubated in dark or under blue light (added **Fig. 2f**). We detected the peak around 450 nm from the dark-treated samples and loss of this peak after exposure to blue light for 15 min; this behavior is consistent with published observations on the purified LOV domain part of PtAUREO1a (Herman *et al.*, 2015, *Biochemistry*, DOI: 10.1021/bi501509z). This result demonstrates that FMN was embedded in our purified AUREO1c protein. We have included the description on this result in lines 578-583.

[figure redacted]

Newly added **Fig. 2f**

From Herman *et al.*, 2015, *Biochemistry*

What are the functional implications of the finding that AUREO1c is dimerized in both light and dark conditions?

(3) Thank you for reminding us the lack of implication discussion after presenting this result. The monomer-dimer transition and conformational changes are common mechanisms involved in the activation process of photoreceptors, such as the UVR8 protein in plants. In aureochromes, there have been different models: the xanthophyte VfAUREO1 was found to be monomeric in dark but dimeric in blue light, whereas PtAUREO1a was found to be dimeric under both conditions (Kroth *et al.*, 2017, *Journal of Plant Physiology*, DOI: 10.1016/j.jplph.2017.06.010). However, there must be structural differences between dark-state and lit-state dimers because the chromophore FMN forms adducts with cysteine residues after blue light exposure. We now discuss the similarities between PtAUREO1a and PtAUREO1c, and point out that the structural differences should be investigated in future studies (lines 572-575 and lines 698-600).

What is the role of light intensity in the AUREO1c-dependent gene expression of LHCX2/3? Is the AUREO1c mediated activation of LHCX2/3 an on/off type of response?

(4) This is an interesting question with implications for the understanding of algal physiology, the understanding of how photoreceptors signal high light stress, and future application of AUREO1c as an optogenetic tool. We address this comment by performing additional experiments and making an analogy to high light sensing in *Arabidopsis thaliana*.

[figure redacted]

Adapted from Sakai *et al.*, 2021, *PNAS*

In *A. thaliana*, PHOT1 and PHOT2 are redundant in regulating hypocotyl curvature (Sakai *et al.*, 2021, *PNAS*, DOI: 10.1073/pnas.101137598). By comparing the curvature response in the single *phot2* (*npl1*) mutant to the *phot1 phot2* double mutant, researchers deduced the light intensity range of PHOT1 function, and found it to saturate around 10 $\mu\text{mol photons m}^{-2} \text{s}^{-1}$ (see the figure above from the reference). Comparison of the single *phot1* (*nph1*) mutant with the double mutant revealed that PHOT2's function is limited under 10 $\mu\text{mol photons m}^{-2} \text{s}^{-1}$ but enhanced under 100 $\mu\text{mol photons m}^{-2} \text{s}^{-1}$. This ability of PHOT2 to tell between low light and higher light is well with its physiological function to respond to high light stress and mediate the chloroplast light avoidance movements, preventing plants from light damages (see the figure above from the reference).

Newly added **Supplementary Fig. 12**

During the revision, we have performed qPCRs to assess the expression of *LHCX2/3* in cells that were initially grown under normal growth conditions and subsequently treated by different intensities of light for 15 min. As shown in the newly added **Supplementary Fig. 12**, the expression of *LHCX2/3* can be induced from normal light (growth light) to higher intensities of light, and this induction is dependent on AUREO1c. In addition, it appears that AUREO1c-mediated activation of *LHCX2/3* is continuously increasing within the range of light intensities that we tested, instead of saturating at a low intensity. This does suggest that AUREO1c has a potential to serve as a sensor of high light stress. This result has

been incorporated and discussed in lines 444-448.

Are other factors (e.g. redox or ROS signals from the chloroplast) come to play?

(5) Redox or ROS signals play crucial roles in various aspects of growth, development, and stress response in photosynthetic organisms. Previous research has often employed pharmacological approaches to test the functions of the chloroplast electron transport chain (ETC) or ROS. During the revision, we performed RT-qPCR to investigate the impact of DCMU, an ETC inhibitor, and N,N'-Dimethylthiourea (DMTU), a ROS scavenger, but obtained results that were difficult to disentangle.

Figure R1 for reviewers

As shown in **Figure R1** above, we found that under high light (HL), DMTU apparently increased the expression of *LHCX3* expression in wild type, but not *LHCX2* expression. DCMU increased the expression of both genes in wild type. This is opposite to its effect on *LHCSR3* in *C. reinhardtii* (Petroustos *et al.*, 2016, *Nature*, DOI: 10.1038/nature19358) but consistent with a previous study in *P. tricornutum* (Hao *et al.*, 2018, *AMB Express*, DOI: 10.1186/s13568-018-0703-3). However, under HL, in the *aureo1c-1* mutant, DCMU increased the expression of *LHCX3* but not *LHCX2*. Given this complication, we feel it will take substantial amount of time to figure out whether the effect of DCMU is independent from AUREO1c or not. Thus, we consider it out of the scope of the current study and acknowledge the limitation of our study. We now add the possibility that these other factors may "interact with the AUREO1c pathway" (lines 903-905). We further mention the recent finding that intracellular CO₂ concentration can work as a signal to regulate photoprotective gene expression in green algae and this could be conserved in diatoms (lines 900-903). Moreover, by performing experiments requested in your next comment, we noticed an upregulation of *LHCX2/3* during

dark-to-light transitions and found this activation to be partially dependent on AUREO1c. By comparing it with previous research, we discuss the cryptochrome CPF1 as another player in *LHCX* expression (lines 896-900).

All data presented here compare normal light to high light conditions, conditions in which AUREO1c is active. What happens when dark-acclimated cells are shifted to low, normal or high light? Would the expression levels of *LHCX2/3* be similar across all light intensities?

(6) Thanks for your suggestion. As shown in response (4), by exploring *LHCX2/3* expression at different light intensities, we clarify that AUREO1c is not an ON/OFF switch. Instead, its activity is stronger when light intensity is higher. The transition from dark is a physiologically relevant process because it occurs every dawn. We performed an additional experiment to compare *LHCX2/3* expression in normal growth light-acclimated cells ("GL"), cells switched from growth light to high light, dark-acclimated cells ("Dark" samples), and cells switched from dark to growth light or high light (newly added **Supplementary Fig. 13**).

Newly added **Supplementary Fig. 13**

We first found that *LHCX2/3* can be upregulated by high light or even growth light exposure after acclimation in dark; this is partly dependent on AUREO1c because the *aureo1c-1* mutant was lower in the expression of both genes than wild type ("Dark to GL 10 min" and "Dark to HL 10 min" samples). So AUREO1c may also play a role during the dawn. These behaviors of *P. tricornutum* wild-type cells, and the influence of AUREO1c are overall similar to *C. reinhardtii* cells and CrPHOT1 (Redekop *et al.*, 2022, *Science Advances*, DOI: 10.1126/sciadv.abn1832). Secondly, there must be AUREO1c-independent factors, and their function seem particularly important when dark-acclimated cells are

exposed to light, because the *aureo1c-1* mutant also showed 5-20x higher levels of *LHCX2/3* expression after this exposure. This is an important finding since our last submission. Thirdly, cells show higher *LHCX2/3* expression at 10 min after the switch from dark to GL than cells acclimated to GL. It was previously shown in both *C. reinhardtii* and *P. tricornutum* that *LHCSR/LHCX* genes would drop in transcript abundance ("relaxation" process) after a period in light treatments so our results are consistent with previous reports. We discuss these results and implications in lines 480-493.

Fig. 4: The authors set out to compare the receptor-to-nucleus signal transmission that controls photoprotective responses in *Chlamydomonas* and *Phaeodactylum*. I find the idea very interesting, but in my opinion the experimental setup would benefit from some improvements, namely that the same trophic mode should be used in both *Chlamydomonas* and *Phaeodactylum* experiments, since acetate that was present in *Chlamydomonas* experiments, is a suppressor of *LHCSR3*. Even though *LHCSR3* mRNA accumulation does occur even in the presence of acetate, when transitioning from low to high light, the levels are 10-20 times lower as compared to phototrophic conditions. As a result, mRNA quantification in these conditions of low transcript abundance, may suffer from insufficient resolution to properly address the impact of HL in the very first minutes of the exposure. I strongly recommend repeating the *Chlamydomonas* experiment using phototrophic medium. The authors may also want to consider starting from darkness instead from NL, as I also mentioned in my previous comment.

(7) Thanks a lot for your suggestion. Indeed, both the acetate-containing TAP and minimal high-salt (HS) media have been used for *LHCSR3* expression studies. During the revision, we have added the HS samples, shown in **Fig. 4**. We moved our original results from TAP medium cultures into **Supplementary Figs. 17, 19, 21**. Within 10 min of high light treatment, the fold of induction and the timepoints showing statistically significant induction were similar between TAP (**Supplementary Fig. 17**) and HS (**Fig. 4a**). The similar fold change between the two conditions does not contradict the fact that acetate suppresses *LHCSR3* expression. This is because in our experiments, the same medium was used for growth light (GL) acclimation and the corresponding high light treatment, and the expression level was normalized to T_0 of treatment (cells in GL). Under GL, our HS

samples already showed a higher level of *LHCSR3* expression compared to the TAP samples (**Supplementary Fig. 18**). This is consistent with a recent study (Ruiz-Sola *et al.*, 2023, *Nature Communications*, DOI: 10.1038/s41467-023-37800-6).

We also updated our RT-qPCR and proteomic analysis of *LHCSR3* expression and now include the data obtained from HS cultures in **Fig. 4c-e and Supplementary Fig. 20**. In our experiments, in the HS medium, *LHCSR3* transcript abundance reaches maximum at around 30-60 min of CL treatment and shows relaxation afterwards (**Fig. 4c; Supplementary Fig. 20**). This relaxation was not seen within 120 min of CL treatment in the TAP medium (**Supplementary Fig. 19**). Correspondingly, the expression level of *LHCSR3* in FL was much lower than that in CL for samples taken at 30 min and 60 min in HS, and for samples taken at all three timepoints tested in TAP (lines 751-757; **Fig. 4c; Supplementary Fig. 19**). Considering the relaxation of signaling (Ruiz-Sola *et al.*, 2023; Taddei *et al.* 2016 *J Exp Bot*, DOI: 10.1093/jxb/erw198), we discuss that more data collection timepoints should be included in future studies on the kinetic pattern of algal *L1818* gene induction (lines 887-891).

As mentioned in response (6), we followed your suggestion to perform experiments on dark-acclimated cells and achieved two interesting findings: AUREO1c plays a role in dark-to-light transition; the role of AUREO1c-independent factors in regulating *LHCX2/3* expression is prominent in this process. Because of the latter complication, we still used transition from normal growth light to high light for the kinetics experiments. However, we have adopted this suggestion when we explored AUREO1c function under a series of different intensities of blue light (**Supplementary Fig. 14**) and proposed starting from darkness as future work in the paragraph concerning the factors beyond AUREO1c (lines 905-908).

Newly added **Supplementary Fig. 18**

A final comment on the ecological relevance of the results presented in Fig. 1c, f showing that the *aureo1c* cultures collapse after 2 days at VHL. Is there an observable phenotype (e.g. pigment composition, growth, NPQ) on a time scale of hours, that would be closer to the conditions the cells would experience at their natural habitat?

(8) This is a valuable suggestion. We measured F_v/F_m at multiple timepoints during VHL treatment, and found that the mutant already showed a decreased F_v/F_m at 2 h (**Supplementary Fig. 8**). We performed NPQ measurements after 4 h of VHL treatment, and found the *aureo1c-1* mutant to be lower than wild type (**Supplementary Fig. 9**). These time frames are more comparable to natural conditions. The loss of chlorophylls in the mutant was evident 1 d after VHL treatment, as discussed in response (1). However, at 6 h, that did not occur (**Supplementary Table 1**), suggesting that the low NPQ or F_v/F_m are not secondary consequences of cell bleaching. These results are discussed in lines 250-254.

Newly added **Supplementary Fig. 8**

Newly added **Supplementary Fig. 9**

Minor comment

Fig. 1d: add error pars, p-values

(9) Thanks for your suggestion. We have now performed statistical comparison between wild type, the mutants, and the COMP line and have added the *P* values.

Fig. 1d: Without wanting to deviate the focus of the work from AUREO1c and LHCX2/3, I believe that this manuscript provides an opportunity to get an overview of the roles of the three AUREO isoforms (1a, 1b and 1c) in regulating expression of all four photoprotection genes (LHCX1-4). In this regard I would encourage the authors to add a supplementary Figure with this information.

(10) Thank you for this suggestion. This is a good point. We have added the data as **Supplementary Fig. 10**. The mutations of the three aureochrome-encoding genes did not lead to a major impact on *LHCX4* transcript accumulation (**panel b**). Interestingly, *LHCX1* exhibited a higher expression level in the *aureo1a-1* mutant than wild type under growth light (**panel a**). We then expanded this analysis to include *aureo1a-2* and *aureo1a-3*, independent mutant alleles of AUREO1a (**panel c**). These two did not show a major difference from wild type. It is possible that a second-site mutation in *aureo1a-1* caused the *LHCX1* expression phenotype. These results underscore the importance of obtaining multiple independent mutant lines or performing genetic complementation in the study of a functional gene. These results are discussed in the Results section (lines 255-258), and partly in the legend of **Supplementary Fig. 10** in order not to disrupt the logic flow of the main text.

Newly added **Supplementary Fig. 10**

Reviewer #2:

A rapid transcriptional opto-switch enables diatom acclimation to dynamic stress.

Zhang et al. use CRISPR/Cas9 methods to knock-out activities of AUREO1a,b and c and LHCX2 and LHCX3 to resolve their interplay and function in acclimation to light stress in the diatom *Phaeodactylum tricornutum*. Using conjugation three mutant lines each of AUREO1a, AUREO1b and AUREO1c were produced and tested under defined light intensities and quality conditions. At 550 $\mu\text{mol photons m}^{-2} \text{s}^{-1}$ maximum quantum yields of PSII was reduced for the *aureo1c* mutant and not in *aureo1a* and *1b* pointing to a role in light acclimation. The *aureo1c* showed strongly reduced survival at light intensities of 1200 $\mu\text{mol photons m}^{-2} \text{s}^{-1}$, and the phenotype of *aureo1c-1* line was rescued by complementation with a *aureo1c*-GFP fusion product. Images (stamp

size) indicate that AUREO1c is localized to the nucleus supporting the role as a bZIP/LOV light regulated transcription factor.

Transcriptomics of wt, aureo mutants and a complemented aureo1c mutant showed reduced expression of two light harvesting protein encoding genes LHCX2 and LHCX3 in the aureo1c mutant indicated that they are under control of aureo1c. LHCX proteins has previously been shown to have a role in NPQ and NPQ was reduced in the aureo1c compared to wt at 900 $\mu\text{mol photons m}^{-2} \text{s}^{-1}$.

The data presented point to a model where AUREO1c receive a blue light induced signal activating expression of the LHCX2 and LHCX3 genes which in an not explained mechanism affect NPQ.

The results are very well documented.

Some comments:

Title could include AUREOCHROME opto switch ...

(11) Thanks for your suggestion. We have changed the title to “A rapid aureochrome opto-switch enables diatom acclimation to dynamic light.”

It is previously established that AUREOCHROMES blue light receptors can act as rapid opto switches and that LHCX's are involved in NPQ and acclimation to light conditions. And that they are nucleus localized sensors for high blue light intensities.

The present manuscript provide a further step and a functional analysis of three specific AUREOCHROMES and two LHCX's. Produced results are solid and provide evidence showing that AUREOCHROME1c specifically control LHCX2 and LHCX3 expression levels thereby affecting NPQ in acclimation to (blue)light stress. Protein size exclusion indicate that AUREOCHROMES as many other TFs primarily are dimerized and have some mass variation pointing to modifications at protein level. Complementation of aureo1c support the results.

Thank you for your affirmation of our work.

The results are well presented and the text read well and is easy to follow, with a few exceptions:

Major point:

The documentation of the biallelic mutants of aureo1a, aureo1b and aureo1c should be improved or at least explained better. It is referred to supplementary fig.3 as proof of bi-allelic changes for all 9 mutants of AUROCHROME's and for the 6 mutants of LHCX2 and 3. Only one mutation is specified for each mutated line indicated that all are mono-biallelic? Gene conversion might explain such variants sometimes, but not for five genes at different chromosomal location?

(12) We apologize for the lack of details in our last submission. In our original manuscript, we have explained that *P. tricornutum* has a high frequency of gene conversion (Bulankova *et al.*, 2021, *Current Biology*, DOI: 10.1016/j.cub.2021.05.013). Now we additionally cite an early CRISPR paper in that first observed the frequent occurrence of apparently homozygous repair outcomes (Nymark *et al.*, 2016, *PLOS One*, DOI: 10.1038/srep24951).

Importantly, we clarify that not all the mutants obtained would be homozygous, but we chose the homozygous ones for a reason. As shown in **Figure R2** below, in the first round, we get clones with either wild-type sequence or mixed sequences, but never a pure, unequivocal sequence. This may be because Cas9 digestion worked (in part of the cells within each clone) after the cells have started dividing on the selective agar plates and the clones would be mosaic. We then picked clones showing alterations from wild type and performed sub-cloning. After this round, we could typically get sub-clones with unequivocal sequences. However, some of the sub-clones still yielded mixed sequences. And for some of the genes, we still had sub-clones with a wild-type sequence. We believe that some of the sub-clones with mixed sequences could be heterozygous, with only one allele edited (scenario a), or with both alleles edited into different indels (scenario b). However, there is still chance that these sub-clones could be mosaic, containing wild-type cells (scenario c). In scenarios a and c, we would not expect the sub-clones to show a knockout mutant phenotype. This is why we chose multiple homozygous lines for physiological characterizations. We now clarify this procedure in the Results section (lines 199-202) and the Methods section (lines 972-977, 992-997). We also clarify that when we designed primers, we chose regions free of SNPs to avoid amplifying just one of the two alleles (lines 974-976). And in the case of *aureo1b-1* mutant, as discussed in response (13) below, the

same amplicon contained a unequivocal InDel sequence at the guide region and a retained SNP within it, the latter indicating amplification from both alleles.

Figure R2 for reviewers

One possibility to overcome this is to search for polymorphisms. These genes are relatively small and have polymorphisms that potentially might clarify if it really is bi-allelic mutants.

(13) Thank you for your suggestion, this is of great significance in providing clear evidence that the variant presented in the manuscript is a bi-allelic homozygous mutation. For *AUREO1a*, *AUREO1c*, and *LHCX2*, we performed an additional round of PCRs with an amplicon covering the entire gene. We then design a sequencing primer to probe a single-nucleotide polymorphism (SNP) for each gene, because other types of polymorphisms, such as InDels, would lead to mixed sequencing results difficult to interpret. For *LHCX3*, we did not find an SNP in the gene so it is not included. For *AUREO1b*, there is an SNP close to the guide region that was already covered in our genotyping PCR that was performed to screen for successful knockouts. These results are presented in the newly added **Supplementary Fig. 5**.

Newly added **Supplementary Fig. 5**

We observed the SNP in wild type remaining in some of the mutants, such as *aureo1a-2*, *aureo1b-1*, *aureo1c-3*, and both lines of *lhcx2* mutants. This observation confirms the presence of both alleles, indicating that there were no large deletions in one of the chromosomes. In the case of *aureo1b-1*, this amplicon is the same as shown in **Supplementary Fig. 4** (original Supplementary Fig. 3) showing an unequivocal sequence at the guide site; and the presence of "M" here excludes the possibility that the PCR only amplified one of the two alleles.

Figure R3 for reviewers. White and black dots represent distinct nucleotides.

We did notice that in some of the mutant lines, such as *aureo1a-1*, *aureo1a-3*, *aureo1b-2*, *aureo1b-3*, *aureo1c-1*, and *aureo1c-2*, the polymorphism observed in wild type disappeared in the mutant, leaving only one of the two versions. Interestingly, the version that remained varied between lines (**Supplementary Fig. 5**; schematic in **Figure R3**). This was also found in our previous study where all the mutants showed a pigment phenotype expected from a knockout mutant while becoming homozygous at the SNP sites (**Figure S2** in Bai *et al.*, 2022, *PNAS*, DOI: 10.1073/pnas.2203708119).

[figure redacted]

From Johzuka-Hisatomi *et al.*, 2008, *Nucleic Acids Research*

Previous studies have provided insights into the loss of polymorphisms, assuming that those SNPs were present in the homology arms when gene conversion occurred in the mutants. In a gene-targeting effort in rice, the authors discovered that nucleotide changes within the vector's homologous segments could be effectively incorporated into the matching genomic sequences of the recombinants (Johzuka-Hisatomi *et al.*, 2008, *Nucleic Acids Research*, DOI: 10.1093/nar/gkn451; figure shown above). The paper proposed mismatch repair as a potential mechanism for introducing these base disparities from the vector into the chromosome. In our **Results** section (lines 214-217), we discuss this mechanism as a possible explanation for the observed disappearance of polymorphisms in some mutants.

Minor points:

Suppl. Fig 3:

Indicate what boxes of the gene models represents at a-e.

(14) We now explain in the legends: "Open green boxes denote exons; red triangles indicate target sites for CRISPR/Cas9-mediated mutagenesis. The target sequences are shown above the WT sequence. The protospacer and the protospacer-adjacent motifs or their complementary sequences are labeled with a red line and a red box respectively. PCR was conducted on each strain with primer locations indicated by red arrows (**Supplementary Data 1**)." Below is one example

from the updated figure (now labeled "Supplementary Fig. 4").

The plus/minus strand/orientation of the target gene sequences would be easier to read if it is presented in an uniform style.

(15) In the revised figure, we have replaced the term "target" with "guide" to unequivocally denote the sequence that resides on the same strand and precedes the "NGG" PAM sequence. Now the reference sequence for the sequencing results is consistently the sequence complementary to the guide sequence. In the cases of *LHCX2*, *LHCX3*, *AUREO1a*, and *AUREO1b*, this reference sequence corresponds to the positive strand. However, for *AUREO1c*, the reference sequence aligns with the negative strand, and therefore, its orientation is depicted inversely compared to the others in the figure.

Abstract line32: ...strong blue light in elution profile.. explain?

(16) We apologize for the unclear description. We meant that in size-exclusion chromatography, there is a shift in the elution profile of the protein after strong blue light excitation, suggesting that the protein indeed responds to light in conformation. However, now we have added an experiment to show that *AUREO1c* responds to blue light using a more common way in photoreceptor research: the protein shows an absorption peak around 450 nm (caused by the FMN cofactor) but loses this peak after blue light-caused FMN-cysteine adduct formation (newly added Fig. 2f). This is a common behavior for photoreceptors containing LOV domains. We have

updated this sentence in the abstract as "In addition, it contains a light-oxygen-voltage (LOV) domain and responds to strong blue light in the absorption spectrum." Please see line 35-37.

Newly added Fig. 2f

Li818 vs LHCX. Keep LHCX nomenclature throughout?

(17) Thanks for your suggestion. We now standardize the description as LHCX throughout. Please see examples in lines 37, 148, and 151.

P5 line 105: .. cell survival... Likely true, but no staining/cell numbers?

(18) During the revision, we measured the cell numbers and chlorophyll fluorescence after 1 d of VHL treatment. As depicted in **Supplementary Fig. 7**, the cell number of the *aureo1c-1* mutant was much lower than that of wild type and the COMP strain, despite that we adjusted them to the same concentration before the VHL treatment. Additionally, most of the mutant cells have lost chlorophyll fluorescence based on flow cytometry. These data suggest severe loss of cell fitness in *aureo1c-1* cells. Additionally, we also detected several observable phenotypes on a time scale of hours, such as low NPQ and low F_v/F_m , as shown in response (8). All these newly added data have been involved in our revised manuscript (lines 225-233, 245-254). In addition, we now use "photosynthetic performance" in the title of the section instead of "survival" (line 149).

Newly added **Supplementary Fig. 7**

Why was line *aureo1c-1* chosen?

(19) The choice of *aureo1c-1* was just because it was the first mutant line that we obtained. We then performed complementation of this line and generation of independent lines in parallel. Three lines with different indels for each gene were constructed, and they exhibited similar phenotypes under high light stress (**Fig. 1c, 1f and Supplementary Fig. 4d-f, 6a, 6b**).

Reviewer #3:

The manuscript deals with the demonstration of the role of a specific blue-light photoreceptor, the aureochrome, in the light response of the model diatom microalga *Phaeodactylum tricornutum*. Using a diversity of approaches and techniques, the authors successively show how AUREO1 c is crucial for photoprotective LHCx genes expression and proteins and synthesis, NPQ induction and photosynthesis performance under high light and fluctuating light. They additionally show the nucleus localization of AUREO1 c and how it directly interacts with the promoters of LHCx genes to regulate their expression. By comparing the light-response dynamics of *P. tricornutum* to *Chlamydomonas* in this framework, they finally propose an explanation for the fast induction of photoprotection in diatoms, and how this could be crucial for their ecological success in mixed coastal waters. The data and conclusions are of high importance for understanding the ecological success of diatoms, which is the most

diverse and productive eukaryotic group of microorganisms, in modern oceans. Beyond, it brings original data on the role of photoreceptors in photosynthetic organisms, and essential differences among phylogenetic lineages (the green which dominates the land versus the red which dominates the oceans). In general, this piece of work is remarkable by the number of techniques used and the amount of data. It is clearly written, well presented, especially the Results section follows a logic step-by-step strategy that is particularly pleasant to read. The figures are also nicely organized and well synthesized. It clearly deserves publication in Nature Communications after some revisions. Here are some general and more specific comments to help the authors improve their manuscript.

Thanks a lot for your recognition of our work.

Light:

-it is essential to show the light spectrum of all the light sources used in this work: the white growth light, all types of experimental lights (white, blue, etc.), and PAM-fluorometer, which I guess, used blue light; a good example is Lines 188-189: 'note that strong blue light is a component of HWL', the authors need to show it and to calculate proportions so that the BL component of WL can be directly compared with HBL.

(20) Thank you for pointing this out. Indeed, it is important to provide the details of the light conditions. The Imaging-PAM fluorometer and Dual-PAM fluorometer in our lab are blue-light and red-light versions respectively, and this has now been indicated in the **Methods** section (lines 1069 and 1078). Considering the suggestions you mentioned below, we created **Supplementary Data 2** to illustrate the intensity and quality of light used in each experiment. We also presented the corresponding spectral graphs for each light culture chamber in the newly added **Supplementary Fig. 2**. The specific light intensity values are compared and discussed in the response (21) below.

Newly added **Supplementary Fig. 2**

-the PAR versus PUR equivalences between all light sources (growth and experimental) need to be calculated; for instance lines 177-184 and the use of two HBL1 and HBL2 intensities (and corresponding Methods section), this is essential and it needs to be clarified with real numbers in order to avoid foggy arguments such as 'more comparable' (line 183).

(21) Thank you for your suggestion. The PAR and PUR for all experiments are now summarized in **Supplementary Data 2**. We updated the concerned section (lines 516-524) into: "To understand whether the function of AUREO1c is blue light signal-associated, we compared the transcriptomes of the *aureo1c-1* mutant and wild-type cells under different light input conditions: growth light (GL), high white light (HWL), high red light (HRL), high blue light 1 (HBL1) and high blue light 2 (HBL2) (**Fig. 2b**, **Supplementary Fig. 2**, and **Supplementary Data 2**). HBL2 was similar to HRL and HWL in photosynthetic active radiation (PAR) but higher in photosynthetically usable radiation (PUR) by 70% and 36%, respectively. HBL1 was additionally included to control for the actual photosynthetic energy intake; it was 18% lower than HWL and similar to HRL in PUR (**Fig. 2b**)." The exact values of PAR, PUR and blue light components of PAR for each light condition used in this RNA-Seq experiment are now provided in **Fig. 2b**. In line 536, we remind readers again that HRL and HBL1 are similar in PUR but the WT/*aureo1c-1* distinctions in transcriptomes are more correlated between the HWL and HBL1 conditions than

between the HWL and HRL conditions (Fig. 2d). This suggests that the blue light component, rather than PUR, is the principal factor influencing AUREO1c function.

Updated Fig. 2b

-experimental lights: what is the most difficult to follow are the different light treatments that have been used, especially the intensities and durations of light exposures are never the same from an experiment to another, this is not impeding the conclusions per se but that would be nice if the authors could better present (directly on figures ? in a summary Table ?) these differences.

(22) We apologize about this confusion, which was mainly caused by the limitations of each culture chamber in terms of light intensity ranges or compatibility with time programming. As mentioned in response (20), We now show the light spectrum of each culture chamber in Supplementary Fig. 2 and provide the light conditions (PAR and PUR) for each experiment in Supplementary Data 2.

-instead of using NL-normal light (which means nothing in a photophysiological framework), better use GL for growth light.

(23) Thanks for your suggestion. We have changed NL (normal light) to GL (growth light) throughout the manuscript.

-as stated by the authors all previous works always used low to moderate intensities of blue light when the current study uses 'high' blue light: it would have been ideal for the present work to additionally include data generated under low blue light to get some kind of comparative baseline with the 'high' blue light. If the authors have such data they should include them, if not, they should, in the Discussion, use previous published

works (Kroth et al., and so on) to compare with their data.

(24) Thanks for your valuable suggestion. Our original manuscript qualitatively described "high blue light" and "low light." However, in reality, photoreceptors are quantitatively enhanced in their activity when light intensity increases. They are just different in light sensitivities. In the land plant *A. thaliana*, the phototropin PHOT1 is saturated in its function around $10 \mu\text{mol m}^{-2} \text{s}^{-1}$ of blue light, whereas PHOT2 can distinguish between $10 \mu\text{mol m}^{-2} \text{s}^{-1}$ of blue light and higher light. Relatedly, PHOT2 is employed as a regulator of high light protection. This is now introduced in lines 115-120. Please also see response (4) above. Additionally, as mentioned in response (20-22), we now describe our light settings in a quantitative way.

Newly added **Supplementary Fig. 14**

Secondly, we respond to this question by analyzing light intensity-dependent *LHCX2/3* expression in wild type and the *aureo1c-1* mutant. As discussed in response (4) and response (6) above, *LHCX2/3* expression and the ratio between wild type and the mutant increase with escalating light intensities, and also respond to the transition from dark to low white light (**Supplementary Figs. 12 and 13**). We also performed experiments by shifting dark-acclimated cells to different intensities of blue light (**Supplementary Fig. 14**). Even under $5 \mu\text{mol m}^{-2} \text{s}^{-1}$ of blue light (the lowest that we can set with our chamber), *LHCX2/3* was induced in wild type, consistent with a previous paper using $3.3 \mu\text{mol m}^{-2} \text{s}^{-1}$ of blue light (Coesel *et al.*, 2009, *EMBO Reports*, DOI: 10.1038/embor.2009.59); and the *aureo1c-1* mutant showed deficient induction of *LHCX2/3* from darkness. When the intensity of blue light is higher, the deficiency is more obvious (in terms of fold change from wild

type). Thus, a brief summary is that we could not define a critical light intensity for activating AUREO1c and the minimum intensity of blue light capable of partially activating AUREO1c function still remains to be determined. These results are described in lines 462-479.

These experiments also highlighted the role of additional regulatory factors besides AUREO1c in controlling *LHCX2/3* expression, particularly noticeable when transitioning dark-acclimated *P. tricornutum* cells to light. The Coesel *et al.* 2009 paper suggested that CPF1 may play a role in this process and we now discuss this in lines 898-900.

Abstract:

-Line 20: Diatoms OFTEN outnumber.

(25) We have adopted this suggestion (line 23).

-Line 22: 'are largely elusive', no, this needs to be modulated, there are other places in the text (see comments below) where the authors tend to underestimate previous works, and such statements need to be avoided. Many previous works regarding 'the mechanisms underlying acclimation of diatoms to high light stress' are not elusive at all: they are clear and strong, and for some of them they simply did not go as far as the present work.

(26) Thanks for your suggestion. We corrected the statement by specifically pointing out the gap in regulators and weakening the tone of the sentence, reading: "While the mechanisms underlying their acclimation to high light stress have been extensively characterized, some aspects, such as the identities and operational mechanisms of regulatory factors, are not yet fully clarified" (lines 24-27). Also, we have added citations of related literature. Please also see our responses (31), (34), (36), (45) and (47) below.

-Line 29, 32 and 34: precise these are named LHCx in diatoms.

(27) Thank you for your suggestion. In the revised manuscript, we have been using *LHCX2* and *LHCX3* in later sentences (lines 33, 34 and 37). In line 33-34 where the *LHCX* genes were first mentioned, we explained that they are in the *LI818* family.

-Line 34: expression of SOME LI818 genes: the present work demonstrates such fact

only for 2 over 4 LHCx in Phaeodactylum (and there are up to 17 LHCx isoforms in other diatom and neighbour haptophyte species).

(28) We now improve the accuracy of the sentence by saying "these two genes" (line 39), following the sentence that literally said "*LHCX2* and *LHCX3*" (line 37).

-Line 39: dynamic LIGHT conditions.

(29) We have adopted this suggestion (line 44).

Introduction:

-Line 51: there are more recent references than number 4.

(30) Thank you for helping us improve the reference part. We have changed reference No. 4 into a recent review: Seth, K. *et al.* Bioprospecting of fucoxanthin from diatoms — Challenges and perspectives. *Algal Research* 60, 102475 (2021). Please see line 64.

-Line 54: references 5-7, please do not forget recent and important works and reviews by Falciatore *et al.*

(31) We have added a highly relevant review in this place: Moejes, F. W. *et al.* A systems-wide understanding of photosynthetic acclimation in algae and higher plants. *J Exp Bot* 68, 2667-2681 (2017). Please see line 67. In addition, we have cited additional work from the Falciatore group in other places as references 17, 22, 39, 54, 57, 63.

-Line 55: 'translational potential on food crops', this statement is unclear or incomplete and needs to be better explained.

(32) Thank you for helping us improve the language of this manuscript! We have revised the sentence as "Moreover, diatoms have unique photosynthetic properties, like their superior ability to harness blue-green light common in underwater settings. If integrated into food crops, this could help fill the green gap in their light utilization spectrum." Please see lines 66-69.

-Line 59: references 8,9: these are really old works and since then some new concepts, especially as regards to photoinhibition, were proposed that are included in more recent works.

(33) We have replaced the references with a more recent review paper that discussed light stress-associated damages and recent progress in photoinhibition: Bassi, R. & Dall'Osto, L. Dissipation of Light Energy Absorbed in Excess: The Molecular Mechanisms. *Annu Rev Plant Biol* 72, 47-76 (2021). Please see lines 72-73.

-Line 69: works by Lepetit et al. need to be cited here.

(34) In the original manuscript, we have been citing work from Lepetit *et al.* in other places. We agree that it is good to start citing the work directly related to LHCXs in this place (please see line 91) and have added these papers: Buck, J. M. et al. *Nat Commun* 10, 4167 (2019); Taddei, L. et al. *Plant Physiology* 177, 953-965 (2018); Buck, J. M., Kroth, P. G. & Lepetit, B. *Plant J* 108, 1721-1734 (2021); Buck, J. M., Wünsch, M., Schober, A. F., Kroth, P. G. & Lepetit, B. *Front Plant Sci* 13, 841058 (2022).

-Line 82: 'diatom survival', this is overstated, please modulate; it is not about their survival but about the maintenance of their photosynthetic performance and productivity.

(35) Thank you for your suggestion. We have replaced the term "survival" with "acclimation" in this place (line 126).

-Line 83: 'await to be demonstrated', this is also overstated, please modulate; there are several recent works showing how LHCx are essential in maintaining photosynthetic performance in diatoms, many of them are cited here at other places in the text (Lepetit, Kroth, Falciatore, Ruban, etc.), and I propose the authors to add this one: <https://doi.org/10.3389/fpls.2022.841058>

(36) We have cited related papers here and revised the sentence as: "Additionally, several studies have demonstrated that certain members in the LHCX family are rapidly transcriptionally activated by the onset of high light, in some cases within five minutes; and the importance of LHCX members in diatom NPQ has been demonstrated." Please refer to lines 101-104.

-Line 83 : 'the mechanism by which specific LHCx isoforms are regulated', please precise you refer to their synthesis.

(37) We have now revised the sentence to clarify that we were discussing the synthesis/expression of LHCX isoforms: "However, the mechanisms by which high

light regulates the expression of specific LHCX isoforms remain unknown." Please see lines 104-105.

Results:

-Line 105: 'survival', same remark as above, overstated, please change with what you have effectively measured and show (F_v/F_m), which is 'photosynthetic performance'.

(38) Thank you for your suggestion. We have made two changes accordingly. One, we have revised this headline into: "The photoreceptor AUREO1c is required for *LHCX2/3* gene expression, NPQ development, and maintenance of photosynthetic performance under high light." Two, we have performed more characterizations on cells treated under very high light (VHL). During the first several hours of treatment, the *aureo1c-1* mutant already showed lowered F_v/F_m (newly added **Supplementary Fig. 8**; shown below) and deficiency to induce NPQ (newly added **Supplementary Fig. 9**).

Newly added **Supplementary Fig. 8**

-Line 118: 'lower F_v/F_m can arise', to be modulated in 'lower F_v/F_m could arise' or equivalent; a lower F_v/F_m can be due to many quenching events and when the light intensity is excessive, it is often a mix of qE and qI, qI being itself ill-defined; the only way to conclude on a direct relationship between lower F_v/F_m and PSII photodamage is to proceed with a low light recovery period after the light stress period: the fraction of F_v/F_m that did not recover is equivalent to the fraction of damaged PSII.

(39) Thank you for your explanation. We have revised the sentence accordingly. Please see the changes on line 180.

-Line 132: 'proteins showed fast recovery after blue light excitation', this statement is unclear and needs better explanation.

(40) We apologize for this confusion. We now clarify the complex terminology and the underlying logic both here and in the manuscript. This response also addresses your subsequent comment.

We would like to elucidate certain points that were unclear in our original manuscript: First, it's important to note that photoreceptors exhibit varying degrees of light sensitivity instead of being simplistic ON/OFF switches. Some reach saturation in their function at relatively low light intensities, indicating higher light sensitivity. Conversely, others require higher light intensities to reach saturation, demonstrating lower light sensitivity. Second, the physiological functions of photoreceptors are intricately linked to their respective light sensitivities. Third, photoreceptors display differences in their reversion kinetics. These variances can be effectively measured using spectroscopy in certain laboratories. Within the same photoreceptor family, there exists a negative correlation between light sensitivity and the rate of reversion.

For the first two points, as we explained in responses (4) and (24), between the two phototropins in *A. thaliana*, PHOT1 function reaches saturation at 10 $\mu\text{mol photons m}^{-2} \text{ s}^{-1}$ of blue light in function whereas PHOT2 does not. Consistently, PHOT2 has evolved a unique function in signaling high light stress and mediating the chloroplast movement responses to circumvent excessive excitation. This is now introduced in the updated manuscript as important background (lines 115-122).

For the third point, we use a scheme below (from Ziegler and Möglich, 2015. *Front. Mol. Biosci*, DOI: 10.3389/fmolb.2015.00030) to show a simplistic view of photoreceptor light cycles. For LOV domain-containing proteins, the "dark-adapted" state corresponds to the state where the FMN cofactor is bound by but not covalently linked to the photoreceptor protein, while the "signaling" state means that the FMN has formed an adduct (photoproduct) with the cysteine residues in the photoreceptor. Only proteins in the dark-adapted state present an absorption peak around 450 nm. Therefore, the standard procedure to measure the kinetics of LOV domain reversion includes the illumination of the protein with blue light, which causes the loss of the 450 nm peak, followed by dark incubation of the protein, and monitoring the recovery of absorption at 450 nm. Alternatively, fluorescence, with an emission peak at 490 nm that corresponds to the dark-state protein-FMN complex, could be measured instead of absorption.

[figure redacted]

Figure from Ziegler and Möglich, 2015

From the above scheme, we can see that the faster the recovery is, the more photons would be required to maintain the signaling state. As shown in the figure below, AtPHOT2 recovers faster than AtPHOT1 (Kasahara *et al.*, 2002 *Plant Physiology*, DOI: 10.1104/pp.002410). The authors interpret this as follows: "For both phot2 LOV1+2 constructs, the relatively rapid dark recovery rate will yield steady-state levels of the cysteinyl adduct under continuous illumination lower than for the phot1 LOV1+2 constructs, with their slower regeneration rates. As a consequence, higher fluences are needed to drive phot2 LOV1+2 to the same photostationary equilibrium value as phot1 LOV1+2. This kinetic difference could explain why phot2 mediates responses mainly to higher fluence rates than does phot1."

[figure redacted]

Figure from Kasahara *et al.*, 2002

In *P. tricornutum*, researchers have measured the reversion of AUREO1a and AUREO1c using the aforementioned procedure (Bannister *et al.*, 2019, *Structural Dynamics*, DOI: 10.1063/1.5095063), and found that AUREO1c recovered faster than AUREO1a. This lends credence to the hypothesis that AUREO1c may serve as a physiological sensor for high light intensity, akin to *A. thaliana* PHOT2, as Bannister *et al.* mentioned: "With a time constant of 850 s and a quantum yield of

23%, AUREO1c reveals a faster recovery time and a much lower sensitivity toward light than AUREO1a, pointing to its role as a high light sensor *in vivo*."

[figure redacted]

Figure from Bannister *et al.*, 2019

We have now revised the original sentence into: "Thirdly, photoreceptors exhibit conformational changes upon exposure to light and revert to their original conformation upon exposure to darkness and the reversion kinetics are negatively correlated with their light sensitivity. In an *in vitro* study, recombinant AUREO1c proteins showed faster recovery than AUREO1a proteins after blue light excitation, suggesting that AUREO1c has lower light sensitivity and may play a physiological role in high light sensing." Please see lines 192-198.

-Line 133: 'suggesting', also unclear how the 1st part of the sentence suggests the second part, the whole sentence needs to be re-written.

(41) Please see our response (40) that addresses the last comment together with this one.

-Lines 146-150: showing DES data is not enough, the authors need to show 'real' DD and DT pigment concentrations per Chl a and per cell.

(42) We have included the additional contents of Ddx and Dtx alongside DES in our revised manuscript in the newly added **Supplementary Table 2** that is shown below.

Supplementary Table 2. The content of fucoxanthin (Fx), diadinoxanthin (Ddx) and diatoxanthin (Dtx) after 10-min and 2-day of 900 $\mu\text{mol photons m}^{-2} \text{s}^{-1}$ of high light treatment. Three independent cultures were used for the quantification. Standard deviations are provided.

Samples	10 min			2 d		
	WT	aureo1c-1	COMP	WT	aureo1c-1	COMP
Fx (fg/cell)	599 \pm 28	697 \pm 68	612 \pm 29	129 \pm 12	83 \pm 23	152 \pm 9
Ddx (fg/cell)	90.8 \pm 5.5	98.6 \pm 11.2	94.5 \pm 4.9	41.2 \pm 4.6	7.3 \pm 2.1	68.5 \pm 5.4
Dtx (fg/cell)	6.7 \pm 0.4	8.6 \pm 0.8	7.7 \pm 0.5	34.4 \pm 3.1	16.2 \pm 4.6	25.3 \pm 9.2
Chl a (fg/cell)	624 \pm 22	678 \pm 72	633 \pm 31	99 \pm 11	82 \pm 25	139 \pm 50
Ddx/Chl a	0.223 \pm 0.006	0.223 \pm 0.006	0.229 \pm 0.001	0.637 \pm 0.037	0.136 \pm 0.009	0.810 \pm 0.247
Dtx/Chl a	0.0169 \pm 0.0014	0.0199 \pm 0.0003	0.0193 \pm 0.003	0.548 \pm 0.014	0.312 \pm 0.050	0.287 \pm 0.032
DES	0.071 \pm 0.007	0.082 \pm 0.001	0.078 \pm 0.001	0.463 \pm 0.018	0.695 \pm 0.019	0.272 \pm 0.074

-Line 157: but DES is lower in the COMP line (supplementary Fig 5b), the authors need to explain this intriguing result.

(43) Thanks for noticing this interesting result. We previously observed that the *aureo1c-1* mutant showed a higher DES than wild type and the COMP line but did not discuss it. As mentioned in response (42), we repeated the analysis and observed this result again (2 d in 900 $\mu\text{mol photons m}^{-2} \text{s}^{-1}$ of light as in the original manuscript, **Supplementary Table 2**; additionally 6 h in VHL, **Supplementary Table 1**). This may be a compensatory effect resulting from impaired NPQ. A higher DES has previously been found in the *C. reinhardtii npq4* mutant disrupted in *LHCSR3* genes (**Supplementary Table 1** in Peers *et al.*, 2009, *Nature*, DOI: 10.1038/nature08587). We now discuss this in lines 264-267.

Another observation is that the COMP line was also lower than wild type in DES. This may be caused by the strong EF1 α promoter used to drive *AUREO1c* expression instead of its native promoter. In fact, we also observed higher NPQ (**Fig. 1d**; **Supplementary Fig. 9**) in the COMP line than wild type, which may lead to a lower compensatory induction of xanthophyll de-epoxidation. In the future, we would prefer to use native promoters to drive gene expression for complementation. We now discuss this in lines 301-307.

-Line 318: 'light fluctuations often last 1-2 h in total', this statement is puzzling to say the less, light fluctuations, especially in coastal systems, are permanent and show different frequencies depending on the scale, this needs to be better explain and justify.

(44) We are sorry for this confusing and inadequate description. Indeed, light fluctuations are permanent and can show different frequencies. We were stating

"1-2 h" because there was a sentence in the previously cited reference (Torzillo *et al.* 2021 in *Cultured Microalgae for the Food Industry*, eds T. Lafarga & G. Ación, pp1-48): "it is common for light fluctuations to occur within a range of 1-2 h." Upon closer examination of the cited reference, we found no evidence to support the statement previously made. We have subsequently removed this part of the sentence. It is possible that the drop in *LHCX3* transcript level reflects a transition of NPQ to longer-term protective strategies, such as degradation of the photosynthetic apparatus. However, this has not been proven in our manuscript. Thus, we did not discuss the decrease of *LHCX3* transcript abundance beyond noting the consistency between our data and the Taddei *et al.* 2016 *J Exp Bot* (DOI: 10.1093/jxb/erw198) paper. Please see lines 750-751.

-Part starting Lines 283-284: the FL treatment used here is raw, next time the authors should better use a more refined set-up as the one used in Lepetit *et al.* (ref 32 here) to study the fine dynamics of LHCx synthesis; in general, over this part of the manuscript, former works, such as ref 32, should be used by the authors to compare with their findings, that will likely strengthen their data/conclusions.

(45) Thank you for suggestion. In future research, we will refer to the study conducted by Lepetit *et al.* (2017, *New Phytol*) and give more consideration to the setup of fluctuating light, as we propose in lines 874-875 and 885-887. We have included more comparisons with previous works in our revised manuscript, such as previous work in *P. tricornutum* (lines 468-473) and even *C. reinhardtii* (lines 450-452). Please also see our response (24).

-Lines 324-325: 'for sufficient photoprotection', please show the NPQ data to support such statement.

(46) Thanks for suggesting this experiment to test our model. We have conducted this detection and added in **Fig. 4e**. In *C. reinhardtii*, the NPQ in FL is lower than that in CL, consistent with our prediction. For *P. tricornutum*, intriguingly, the NPQ in FL is higher than that in CL, despite the similar abundance of LHCX2 and LHCX3 proteins (**Fig. 4d**). This suggests that certain factors beyond LHCX proteins may contribute to the difference between the NPQ under FL and CL conditions. This requires further studies to elucidate. We now discuss these new results in lines 793-800.

Newly added Fig. 4e

We did observe that the NPQ curves of our *C. reinhardtii* strain (CC-5325) differ from those in *P. tricornutum* in shape. But this pattern is similar to published results using this strain (Arend *et al.*, 2023, *Nature Communications*, DOI: 10.1038/s41467-023-38183-4) and it has been reported that different *C. reinhardtii* strains can vary significantly in kinetic patterns (Ruiz-Sola *et al.*, 2018, *Methods in Molecular Biology*, DOI: 10.1007/978-1-4939-8654-5_21). We could reproduce the finding that CL induced higher NPQ than FL in two other *C. reinhardtii* strains attempted (Figure R4), which increases our confidence in our conclusion.

Figure R4 for reviewers

Discussion:

The discussion is a bit strange, it is quite short (due to the fact some specific points have been justified/discussed in the Results section) and it reads more like an extended Conclusion. I am personally fine with it as the Results section reads well, but the Editor might feel different about it.

(47) Thanks for the comment. In the first part of the **Discussion** section, we now summarize the results (original and newly added ones) and reason how they allowed us to conclude that "AUREO1c detects blue light signals and triggers high-level NPQ effector gene expression under high light conditions." This should be

more like a typical beginning of the **Discussion** section. Please see lines 826-838.

-Lines 354-360: please cite the Figure 5 already here.

(48) Thanks for your suggestion. During the rewriting process following your above suggestion, we have moved the sentence to lines 859-865, where **Fig. 5** is cited.

Figures:

-Supplementary figure 1: the localization of green microalgae is unclear, and should be better shown: are they present in some kind of sea-water cuvettes or is this a representation of a freshwater system? The boarder between the oceans and the land with their continental aquatic systems (lakes, ponds, rivers, etc.) should be better drawn. Diatom side: the food-web should be better shown too, it is strange that a diatom cell is bigger than a fish. In general, and even if this is a schematic sketch, diatoms and green microalgae should be shown with a smaller size.

(49) Thanks for your suggestion. We were trying to show green microalgae in freshwater systems. In the new **Supplementary Fig. 1**, we have put the water body to the left of land to avoid misleading readers. We have also adjusted the size of fish and algal cells.

Updated **Supplementary Fig. 1**

We thank all Reviewers again for their careful examination of our figure

preparations and writing, in addition to the comments that have greatly helped improve the manuscript!

REVIEWERS' COMMENTS

Reviewer #1 (Remarks to the Author):

I appreciate the effort the authors have made to address all my comments. I very much enjoyed reading the rebuttal letter and the updated manuscript, which I recommend for publication in NCOMMS.

One small final comment: When discussing the relaxation of signalling, the authors may also want to cite Aihara et al. for completeness. <https://doi.org/10.1038/s41477-018-0332-5>

My congratulations to the authors for a very thorough paper!

Reviewer #3 (Remarks to the Author):

The authors really did a great job in answering my comments and, in general, in strongly improving their work and manuscript. I definitely recommend publication.

Johann Lavaud

Responding to

REVIEWER COMMENTS

Reviewer #1

I appreciate the effort the authors have made to address all my comments. I very much enjoyed reading the rebuttal letter and the updated manuscript, which I recommend for publication in NCOMMS.

One small final comment: When discussing the relaxation of signalling, the authors may also want to cite Aihara et al. for completeness. <https://doi.org/10.1038/s41477-018-0332-5>

My congratulations to the authors for a very thorough paper!

(1) Thank you for your recognition and feedback! We have added the reference (ref No. 85) to the corresponding sentence in the Discussion section.

Reviewer #3

The authors really did a great job in answering my comments and, in general, in strongly improving their work and manuscript. I definitely recommend publication.

Johann Lavaud

(2) Thank you for your recognition and feedback!